# Tensor decomposition of higher-order correlations by nonlinear Hebbian plasticity

**Gabriel Koch Ocker**
Department of Mathematics and Statistics
Boston University
Boston, MA 02136
gkocker@bu.edu

**Michael A. Buice**
MindScope Program
Allen Institute
Seattle, WA 98109
michaelbu@alleninstitute.org

## Abstract

Biological synaptic plasticity exhibits nonlinearities that are not accounted for by classic Hebbian learning rules. Here, we introduce a simple family of generalized nonlinear Hebbian learning rules. We study the computations implemented by their dynamics in the simple setting of a neuron receiving feedforward inputs. These nonlinear Hebbian rules allow a neuron to learn tensor decompositions of its higher-order input correlations. The particular input correlation decomposed and the form of the decomposition depend on the location of nonlinearities in the plasticity rule. For simple, biologically motivated parameters, the neuron learns eigenvectors of higher-order input correlation tensors. We prove that tensor eigenvectors are attractors and determine their basins of attraction. We calculate the volume of those basins, showing that the dominant eigenvector has the largest basin of attraction. We then study arbitrary learning rules and find that any learning rule that admits a finite Taylor expansion into the neural input and output also has stable equilibria at generalized eigenvectors of higher-order input correlation tensors. Nonlinearities in synaptic plasticity thus allow a neuron to encode higher-order input correlations in a simple fashion.

## 1 Introduction

In Hebbian learning, potentiation of the net synaptic weight between two neurons is driven by the correlation between pre- and postsynaptic activity [1]. That postulate is a cornerstone of the theory of synaptic plasticity and learning [2, 3]. In its basic form, the Hebbian model leads to runaway potentiation or depression of synapses, since the pre-post correlation increases with increasing synaptic weight [4]. That runaway potentiation can be stabilized by supplemental homeostatic plasticity dynamics [5], by weight dependence in the learning rule [6, 7], or by synaptic scaling regulating a neuron's total synaptic weight [8, 9]. In 1982, Erkki Oja observed that a linear neuron with Hebbian plasticity and synaptic scaling learns the first principal component of its inputs [10]. In 1985, Oja and Karhunen proved that this is a global attractor of the Hebbian dynamics [11]. This led to a fountain of research on unsupervised feature learning in neural networks [12, 13].

Principal component analysis (PCA) describes second-order features of a random variable. Both naturalistic stimuli and neural activity can, however, exhibit higher-order correlations [14, 15]. Canonical models of retinal and thalamic processing whiten inputs, removing pairwise features [16–22]. Beyond-pairwise features, encoded in tensors, can provide a powerful substrate for learning from data [23–27].

The basic Hebbian postulate does not take into account fundamental nonlinear aspects of biological synaptic plasticity in cortical pyramidal neurons. First, synaptic plasticity depends on beyond-pairwise activity correlations [28–33]. Second, spatially clustered and temporally coactive synapses exhibit

correlated and cooperative plasticity [34–41]. There is a rich literature on computationally motivated forms of nonlinear Hebbian learning (section 3.2). Here, we will prove that these biologically motivated nonlinearities allow a neuron to learn higher-order features of its inputs.

We study the dynamics of a simple family of generalized Hebbian learning rules, combined with synaptic scaling (eq. 1). Equilibria of these learning rules are invariants of higher-order input correlation tensors. The order of input correlation (pair, triplet, etc.) depends on the pre- and postsynaptic nonlinearities of the learning rule. When the only nonlinearity in the plasticity rule is postsynaptic, the steady states are eigenvectors of higher-order input correlation tensors [42, 43]. We prove that these eigenvectors are attractors of the generalized Hebbian plasticity dynamics and characterize their basins of attraction.

Then, we study further generalizations of these learning rules. We show that any plasticity model (with a finite Taylor expansion in the synaptic input, neural output, and synaptic weight) has steady states that generalize those tensor decompositions to multiple input correlations, including generalized tensor eigenvectors. We show that these generalized tensor eigenvectors are stable equilibria of the learning dynamics. Due to the complexity of the arbitrary learning rules, we are unable to fully determine their basins of attraction. We do find that these generalized tensor eigenvectors are in an attracting set for the dynamics, and characterize its basin of attraction. Finally, we conclude by discussing extensions of these results to spiking models and weight-dependent plasticity.

## 2 Results

Take a neuron receiving $K$ inputs $\mathrm{x}_i(t)$, $i \in [K]$, each filtered through a connection with synaptic weight $J_i(t)$ to produce activity $n(t)$. We consider synaptic plasticity where the evolution of $J_i$ can depend nonlinearly on the postsynaptic activity $n(t)$, the local input $\mathrm{x}_i(t)$, and the current synaptic weight $J_i(t)$. We model these dependencies in a learning rule $f$:

$$J_i(t + dt) = \frac{J_i(t) + {}^{dt}/\tau\, f_i(n(t), \mathrm{x}_i(t), J_i(t))}{\left\| \boldsymbol{J}(t) + {}^{dt}/\tau\, \boldsymbol{f}\big(n(t), \mathbf{x}(t), \boldsymbol{J}(t)\big) \right\|}, \text{ where } f_i(t) = n^a(t)\, \mathrm{x}_i^b(t)\, J_i^c(t). \tag{1}$$

The parameter $a$ sets the output-dependent nonlinearity of the learning rule, $b$ sets the input-dependent nonlinearity, and $c$ sets its dependence on the current synaptic weight. Eq. 1 assumes a simple form for these nonlinearities; we discuss arbitrary nonlinear learning rules in section 2.3. We assume that $a$ and $b$ are positive integers, as in higher-order voltage or spike timing–dependent plasticity (STDP) models [44–46]. The scaling by the norm of the synaptic weight vector, $\|\boldsymbol{J}\|$, models homeostatic synaptic scaling [8–10]. Bold type indicates a vector, matrix, or tensor (depending on the variable) and regular font with lower indices indicates elements thereof. Roman type denotes a random variable ($\mathbf{x}$).

We assume that $\mathbf{x}(t)$ is drawn from a stationary distribution with finite moments of order $a + b$. Combined with a linear neuron, $n = \boldsymbol{J}^T \mathbf{x}$, and a slow learning rate, $\tau \gg dt$, this implies the following dynamics for $\boldsymbol{J}$ (appendix A.1):

$$\tau \dot{J}_i = J_i^c \sum_\alpha \mu_{i,\alpha}(\boldsymbol{J}^{\otimes a})_\alpha - J_i \sum_{j,\alpha} J_j^{c+1} \mu_{j,\alpha}(\boldsymbol{J}^{\otimes a})_\alpha. \tag{2}$$

In eq. 2, $\dot{J}_i = dJ_i/dt$, $\alpha = (j_1, \ldots, j_a)$ is a multi-index, and $\otimes$ is the vector outer product; $\boldsymbol{J}^{\otimes a}$ is the $a$-fold outer product of the synaptic weight vector $\boldsymbol{J}$. $\boldsymbol{\mu}$ is a higher-order moment (correlation) tensor of the inputs:

$$\mu_{i,\alpha} = \langle \mathrm{x}_i^b (\mathbf{x}^{\otimes a})_\alpha \rangle_{\mathbf{x}} \tag{3}$$

where $\langle \rangle_{\mathbf{x}}$ denotes the expectation with respect to the distribution of the inputs. $\boldsymbol{\mu}$ is an $(a+1)$-order tensor containing an $(a+b)$-order joint moment of $\mathbf{x}$. The order of the tensor refers to its number of indices, so a vector is a first-order tensor and a matrix a second-order tensor. $\boldsymbol{\mu}$ is cubical; each mode of $\boldsymbol{\mu}$ has the same dimension $K$. In the first term of eq. 2, for example, $\sum_\alpha \mu_{i,\alpha}(\boldsymbol{J}^{\otimes a})_\alpha$ takes the dot product of $\boldsymbol{J}$ along modes 2 through $a + 1$ of $\boldsymbol{\mu}$.

### 2.1 Steady states of nonlinear Hebbian learning

If we take $a = b = 1$, then $\boldsymbol{\mu}$ is the second-order correlation of $\mathbf{x}$ and $\alpha$ is just the index $j$. With $c = 0$ also, eq. 2 reduces to Oja's rule and $\boldsymbol{J}$ is guaranteed to converge to the dominant eigenvector

of $\boldsymbol{\mu}$ [10, 11]. We next investigate the steady states of eq. 2 for arbitrary $(a, b) \in \mathbb{Z}_+^2$, $c \in \mathbb{R}$. $J_i = 0$ is a trivial steady state. At steady states of eq. 2 where $J_i \neq 0$,

$$\sum_\alpha \mu_{i,\alpha}(\boldsymbol{J}^{\otimes a})_\alpha = \lambda J_i^{1-c}, \text{ where } \lambda(\boldsymbol{\mu}, \boldsymbol{J}) = \sum_{j,\alpha} J_j^{c+1} \mu_{j,\alpha}(\boldsymbol{J}^{\otimes a})_\alpha, \tag{4}$$

so that $\boldsymbol{J}$ is invariant under the multilinear map of $\boldsymbol{\mu}$ except for a scaling by $\lambda$ and element-wise exponentiation by $1 - c$. For two parameter families $(a, b, c)$, eq. 4 reduces to different types of tensor eigenequation [47, 42, 43]. We next briefly describe these and some of their properties.

First, if $a + c = 1$, we have the tensor eigenequation $\sum_\alpha \mu_{i,\alpha}(\boldsymbol{J}^{\otimes a})_\alpha = \lambda J_i^a$. Qi called $\lambda$, $\boldsymbol{J}$ the tensor eigenpair [42] and Lim called them the $\ell^a$-norm eigenpair [43]. There are $Ka^{K-1}$ such eigenpairs [42]. If $\boldsymbol{\mu} \geq 0$ element-wise, then it has a unique largest eigenvalue with a real, non-negative eigenvector $\boldsymbol{J}$, analogous to the Perron-Frobenius theorem for matrices [43, 48]. If $\boldsymbol{\mu}$ is weakly irreducible, that eigenvector is strictly positive [49]. In contrast to matrix eigenvectors, however, for $a > 1$ these tensor eigenvectors are not necessarily invariant under orthogonal transformations [42].

If $c = 0$, we have another variant of tensor eigenvalue/vector equation:

$$\sum_\alpha \mu_{i,\alpha}(\boldsymbol{J}^{\otimes a})_\alpha = \lambda J_i \tag{5}$$

Qi called these $\lambda$, $\boldsymbol{J}$ an E-eigenpair [42] and Lim called them the $\ell^2$-eigenpair [43]. In general, a tensor may have infinitely many such eigenpairs. If the spectrum of a $K$-dimensional tensor of order $a + 1$ is finite, however, there are $(a^K - 1)/(a-1)$ eigenvalues counted with multiplicity, and the spectrum of a symmetric tensor is finite [50, 51]. (If $b = 1$, $\boldsymbol{\mu}$ is symmetric.) Unlike the steady states when $a + c = 1$, these eigenpairs are invariant under orthogonal transformations [42]. For non-negative $\boldsymbol{\mu}$, there exists a positive eigenpair [52]. It may not be unique, however, unlike the largest eigenpair for $a + c = 1$ (an anti-Perron-Frobenius result) [51]. In the remainder of this paper we will usually focus on parameter sets with $c = 0$ and use "tensor eigenvector" to refer to those of eq. 5.

## 2.2 Dynamics of nonlinear Hebbian learning

For the linear Hebbian rule, $(a, b, c) = (1, 1, 0)$, Oja and Karhunen proved that the first principal component of the inputs is a global attractor of eq. 2 [11]. We thus asked whether the first tensor eigenvector is a global attractor of eq. 2 when $c = 0$ but $(a, b) \neq (1, 1)$. We first simulated the nonlinear Hebbian dynamics. For the inputs $\mathbf{x}$, we whitened $35 \times 35$ pixel image patches sampled from the Berkeley segmentation dataset (fig. 1a; [53]). For $b \neq 1$, the correlation of these image patches was not symmetric (fig. 1b). The mean squared error of the canonical polyadic (CP) approximation of these tensors was higher for $b = 1$ than $b = 2$ (fig. 1d). It decreased slowly past rank $\sim 10$, and the rank of the input correlation tensors was at least 30 (fig. 1d).

The nonlinear Hebbian learning dynamics converged to an equilibrium from random initial conditions (e.g., fig. 1e, f), around which the weights fluctuated due to the finite learning timescale $\tau$. Any equilibrium is guaranteed to be some eigenvector of the input correlation tensor $\boldsymbol{\mu}$ (section 2.1). For individual realizations of the weight dynamics, we computed the overlap between the final synaptic weight vector and each of the first 10 eigenvectors (components of the Tucker decomposition) of the corresponding input correlation $\boldsymbol{\mu}$ [47, 54]. The dynamics most frequently converged to the first eigenvector. For a non-negligible fraction of initial conditions, however, the nonlinear Hebbian rule converged to subdominant eigenvectors (fig. 1g,h). The input correlations $\boldsymbol{\mu}$ did have a unique dominant eigenvector (fig. 3a, blue), but the dynamics of eq. 2 did not always converge to it. This finding stands in contrast to the standard Hebbian rule, which must converge to the first eigenvector if it is unique [11].

While the top eigenvector of a matrix can be computed efficiently, computing the top eigenvector of a tensor is, in general, NP-hard [55]. To understand the learning dynamics further, we examined them analytically. Our main finding is that with $(b, c) = (1, 0)$ in the generalized Hebbian rule, eigenvectors of $\boldsymbol{\mu}$ are attractors of eq. 2. Contrary to the case when $a = 1$ (Oja's rule), the dynamics are thus *not* guaranteed to converge to the first eigenvector of the input correlation tensor when $a > 1$. The first eigenvector of $\boldsymbol{\mu}$ does, however, have the largest basin of attraction.

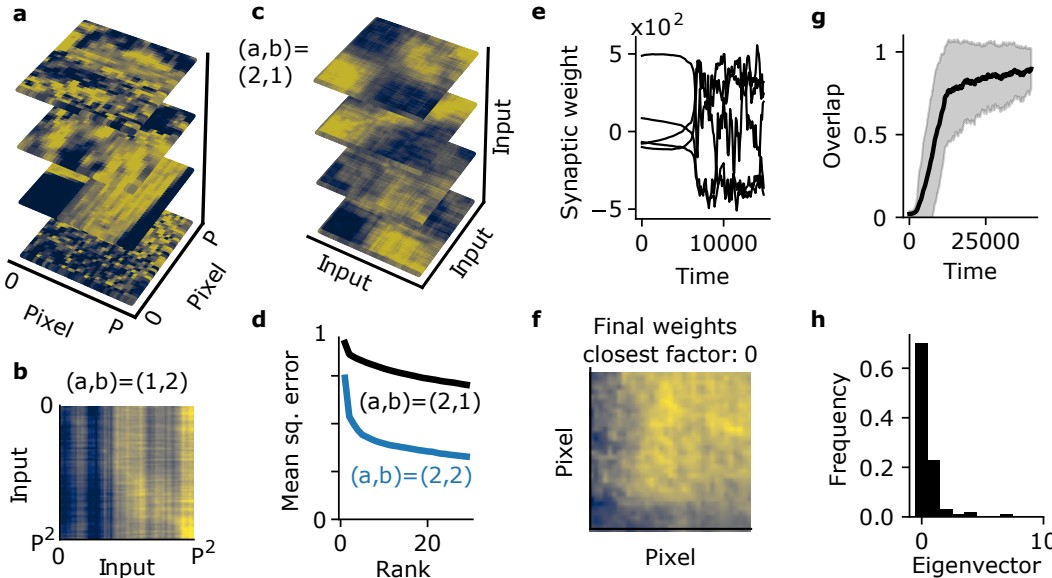

Figure 1: Convergence of higher-order Hebbian plasticity to tensor eigenvectors of natural image correlations. **a)** Example $P \times P$ image patches ($P = 35$ pixels). **b)** Third-order input correlation $\boldsymbol{\mu}$ that drives plasticity for $a = 1, b = 2$. **c)** Third-order input correlation $\boldsymbol{\mu}$ that drives plasticity for $a = 2, b = 1$. **d)** Mean squared error of the CP approximation of the input correlations. (Curves: mean across 4 initializations for the alternating least-squares computation of the CP, with standard deviation within the line thickness.) **e)** Learning dynamics on natural inputs: five randomly selected synaptic weights with $(a, b, c) = (2, 1, 0)$. **f)** Example final synaptic weight configuration with $(a, b, c) = (2, 1, 0)$. **g)** Overlap of the synaptic weights with their final closest singular vector, $\boldsymbol{U}_i^T \boldsymbol{J}$. Solid line: mean over 10 samples of natural image patches and 10 realizations of the weight dynamics for each input patch. Shaded region: standard deviation. Initial conditions for $\boldsymbol{J}$ are chosen uniformly on the unit sphere. **h)** Histogram of the closest singular vector to the final synaptic weights for $(a, b, c) = (2, 1, 0)$.

**Theorem 1.** *In eq. 2, take $(b, c) = (1, 0)$. Let $\boldsymbol{\mu}$ be a cubical, symmetric tensor of order $a + 1$ and odeco with $R$ components:*

$$\boldsymbol{\mu} = \sum_{r=1}^{R} \lambda_r \left( \boldsymbol{U}_r \right)^{\otimes a+1} \tag{6}$$

*where $\boldsymbol{U}$ is a matrix of unit-norm orthogonal eigenvectors: $\boldsymbol{U}^T \boldsymbol{U} = \boldsymbol{I}$. Let $\lambda_i > 0$ for each $i \in [R]$ and $\lambda_i \neq \lambda_j \ \forall \ (i, j) \in [R] \times [R]$ with $i \neq j$. Then for each $k \in [R]$:*

1. *With any odd $a > 1$, $\boldsymbol{J} = \pm \boldsymbol{U}_k$ are attracting fixed points of eq. 2 and their basin of attraction is $\bigcap_{i \in [R] \setminus k} \left\{ \boldsymbol{J} : \left| \boldsymbol{U}_i^T \boldsymbol{J} / \boldsymbol{U}_k^T \boldsymbol{J} \right| < \left( \lambda_k / \lambda_i \right)^{1/(a-1)} \right\}$. Within that region, the separatrix of $+\boldsymbol{U}_k$ and $-\boldsymbol{U}_k$ is the hyperplane orthogonal to $\boldsymbol{U}_k^T$: $\left\{ \boldsymbol{J} : \boldsymbol{U}_k^T \boldsymbol{J} = 0 \right\}$.*

2. *With any even positive $a$, $\boldsymbol{J} = \boldsymbol{U}_k$ is an attracting fixed point of eq. 2 and its basin of attraction is $\left\{ \boldsymbol{J} : \boldsymbol{U}_k^T \boldsymbol{J} > 0 \right\} \bigcap_{i \in [R] \setminus k} \left\{ \boldsymbol{J} : \boldsymbol{U}_i^T \boldsymbol{J} / \boldsymbol{U}_k^T \boldsymbol{J} < \left( \lambda_k / \lambda_i \right)^{1/(a-1)} \right\}$.*

3. *With any even positive $a$, $\boldsymbol{J} = \boldsymbol{0}$ is a neutrally stable fixed point of eq. 2 with basin of attraction $\left\{ \boldsymbol{J} : \sum_{j=1}^{R} (\boldsymbol{U}_j^T \boldsymbol{J})^2 < 1 \wedge \boldsymbol{U}_k^T \boldsymbol{J} < 0 \ \forall \ k \in [R] \right\}$.*

**Remark.** *Each component of the orthogonal decomposition of an odeco tensor $\boldsymbol{\mu}$ (eq. 5) is an eigenvector of $\boldsymbol{\mu}$. If $R < (a^K - 1)/(a-1)$, there are additional eigenvectors. The components of the orthogonal decomposition are the* robust *eigenvectors of $\boldsymbol{\mu}$: the attractors of its multilinear map [26, 56]. The non-robust eigenvectors of an odeco tensor are fixed by its robust eigenvectors and their eigenvalues [57].*

The proof of theorem 1 is given in appendix A.2. To prove theorem 1, we project $\boldsymbol{J}$ onto the eigenvectors of $\boldsymbol{\mu}$, and study the dynamics of the loadings $\boldsymbol{v} = \boldsymbol{U}^T \boldsymbol{J}$. This leads to the discovery of a collection of unstable manifolds: each pair of axes $(i, k)$ has an associated unstable hyperplane $v_i = v_k \left(\lambda_k / \lambda_i\right)^{1/(a-1)}$ (and if $a$ is odd, also the corresponding hyperplane with negative slope). These partition the phase space into the basins of attraction of the eigenvectors of $\boldsymbol{\mu}$.

For example, consider a fourth-order input correlation (corresponding to $a = 3$ in eq. 1) with two eigenvectors. The phase portrait of the loadings is in fig. 2a, with the nullclines in black and attracting and unstable manifolds in blue. The attracting manifold is the unit sphere. There are two unstable hyperplanes that partition the phase space into the basins of attraction of $(0, \pm 1)$ and $(\pm 1, 0)$, where the synaptic weights $\boldsymbol{J}$ are an eigenvector of $\boldsymbol{\mu}$.

For even $a$, only the unstable hyperplanes with positive slope survive (fig. 2b, blue line). The unit sphere (fig. 2b, blue) is attracting from any region where at least one loading is positive. Its vertices $[1] \times [1]$ are equilibria; $(v_1, v_2) = (1, 0)$ and $(0, 1)$ are attractors and the unstable hyperplane separates their basins of attraction. For the region with all loadings negative that is the basin of attraction of the origin, noise will drive the system away from zero towards one of the eigenvector equilibria.

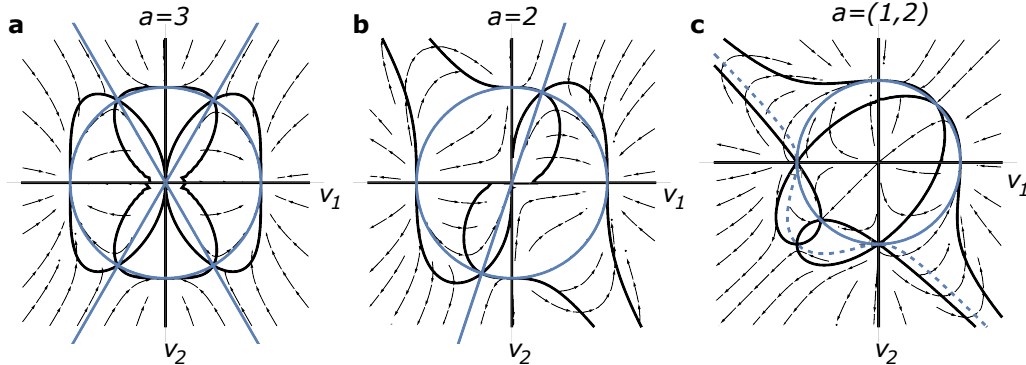

Figure 2: Example dynamics of the loadings, $\boldsymbol{v} = \boldsymbol{U}^T \boldsymbol{J}$, for rank-two input correlations. Black curves: nullclines of $v_1, v_2$. Blue curves: stable sets and separatrices of the basins of attraction of the sparse attractors for $\boldsymbol{v}$. **a)** Odd $a$ ($a = 3$). The stable set is the unit sphere, and the separatrices are $v_2 = \pm v_1 \left(\lambda_1 / \lambda_2\right)^{1/a-1}$. **b)** Even $a$ ($a = 2$). The unstable set is $v_2 = v_1 \left(\lambda_1 / \lambda_2\right)^{1/a-1}$ (solid blue line). In **(a,b)**, $(\lambda_1, \lambda_2) = (3, 1)$. **c)** Phase portrait for a two-term learning rule (eq. 9). All parameters of the input correlation tensors $(\lambda_{mr}, A_i)$ are equal to one. Dashed blue curve: $\mathcal{L} = \{\boldsymbol{v} : \sum_{m,j} \lambda_{mj} v_j^{a_m+1} = 0\}$.

By partitioning the phase space of $\boldsymbol{J}$ into basins of attraction for eigenvectors of $\boldsymbol{\mu}$, theorem 1 also allows us to determine the volumes of those basins of attraction. The basins of attraction are open sections of $\mathbb{R}^K$ so we measure their volume relative to that of a large hypercube.

**Corollary 1.1.** *Let $V_k$ be the relative volume of the basin of attraction for $\boldsymbol{J} = \boldsymbol{U}_k$. For odd $a > 1$,*

$$V_k = R^{-1} \prod_{i=1}^{R} \left(\frac{\lambda_k}{\lambda_i}\right)^{1/(a-1)} \tag{7}$$

**Corollary 1.2.** *Let $V_k$ be the relative volume of the basin of attraction for $\boldsymbol{J} = \boldsymbol{U}_k$. For even positive $a$,*

$$
\begin{aligned}
V_k = 2^{1-R} \Bigg( & R^{-1} \prod_{i} \left(\frac{\lambda_k}{\lambda_i}\right)^{1/(a-1)} + (R-1)^{-1} \sum_{j \neq k} \prod_{i \neq j} \left(\frac{\lambda_k}{\lambda_i}\right)^{1/(a-1)} \\
& + (R-2)^{-1} \sum_{j,l \neq k} \prod_{i \neq j,l} \left(\frac{\lambda_k}{\lambda_i}\right)^{1/(a-1)} + \ldots + 1 \Bigg)
\end{aligned}
\tag{8}
$$

The calculations for corollaries 1.1, 1.2 are given in appendix A.2. We see that the volumes of the basins of attraction depend on the spectrum of $\boldsymbol{\mu}$, its rank $R$, and its order $a$. The result for odd $a$ also

provides a lower bound on the volume for even $a$. The result is simpler for odd $a$ so we focus our discussion here on that case.

While eigenvectors of $\boldsymbol{\mu}$ with small eigenvalues contribute little to values of the input correlation $\boldsymbol{\mu}$, they can have a large impact on the basins of attraction. The volume of the basin of attraction of eigenvector $k$ is proportional to $\lambda_k^{R/(a-1)}$. An eigenvector with eigenvalue $\epsilon$ scales the basins of attraction for the other eigenvectors by $\epsilon^{-1/a-1}$. The relative volume of two eigenvectors' basins of attraction is, however, unaffected by the other eigenvalues whatever their amplitude. With $a$ odd, the ratio of the volumes of the basins of attraction of eigenvectors $k$ and $j$ is $V_k/V_j = \left(\lambda_k/\lambda_j\right)^{R/a-1}$.

We see in theorem 1 that attractors of eq. 2 are points on the unit hypersphere, $\mathcal{S}$. For odd $a$, $\mathcal{S}$ is an attracting set for eq. 2. For even $a$, the section of $\mathcal{S}$ with at least one positive coordinate is an attracting set (fig. 2a, b, blue circle; see proof of theorem 1 in appendix A.2). We thus next computed the surface area of the section of $\mathcal{S}$ in the basin of attraction for eigenvector $k$, $A_k$ (corollary 1.3 in appendix A.2). The result requires knowledge of all non-negligible eigenvalues of $\boldsymbol{\mu}$, and the ratio $A_k/A_j$ does not exhibit the cancellation that $V_k/V_j$ does for odd $a$. We saw in simulations with natural image patch inputs and initial conditions for $\boldsymbol{J}$ chosen uniformly at random on $\mathcal{S}$, the basin of attraction for $\boldsymbol{U}_1^T$ was $\sim 3\times$ larger than that for $\boldsymbol{U}_2^T$ and the higher eigenvectors had negligible basins of attraction (fig. 1h).

In Oja's model, $(a, b, c) = (1, 1, 0)$ in eq. 1, if the largest eigenvalue has multiplicity $d > 1$ then the $d$-sphere spanned by those codominant eigenvectors is a globally attracting equilibrium manifold for the synaptic weights. The corresponding result for $a > 1$ is (for the formal statement and proof, see corollary 1.4 in appendix A.2):

**Corollary 1.3.** *(Informal) If any $d$ robust eigenvalues of $\boldsymbol{\mu}$ are equal, the $d$-sphere spanned by their robust eigenvectors is an attracting equilibrium manifold and its basin of attraction is defined by each of those $d$ eigenvectors' basins of attraction boundaries with the other $R - d$ robust eigenvectors.*

### 2.3 Arbitrary learning rules

So far we have studied phenomenological plasticity rules in the particular form of eq. 1. The neural output $n$, input $x_i$, and synaptic weight $J_i$ were each raised to a power and then multiplied together. Changes in the strength of actual synapses are governed by complex biochemical, transcriptional, and regulatory pathways [58]. We view these as specifying some unknown function of the neural output, input, and synaptic weight, $\boldsymbol{f}(n, \mathbf{x}, \boldsymbol{J})$. That function might not have the form of eq. 1. So, we next investigate the dynamics induced by arbitrary learning rules $\boldsymbol{f}$. We see here that under a mild condition, any equilibrium of the plasticity dynamics will have a similar form as the steady states of eq. 2. If $f$ does not depend on $\boldsymbol{J}$ except through $n$, equilibria will be generalized eigenvectors of higher-order input correlations.

The Taylor expansion of $\boldsymbol{f}$ around zero is:

$$f_i\left(n(t), \mathrm{x}_i(t), J_i(t)\right) = \sum_{m=1}^{\infty} A_m\, n^{a_m}(t)\, \mathrm{x}_i^{b_m}(t)\, J_i^{c_m}(t) \tag{9}$$

where the coefficients $A_m$ are partial derivates of $\boldsymbol{f}$. We assume that there exists a finite integer $N$ such that derivatives of order $N + 1$ and higher are negligible compared to the lower-order derivatives. We then approximate $\boldsymbol{f}$, truncating its expansion after those $N$ terms. With a linear neuron, synaptic scaling, and slow learning, this implies the plasticity dynamics

$$\tau \dot{J}_i = \sum_m J_i^{c_m} \sum_{\alpha_m} m\boldsymbol{\mu}_{i,\alpha_m} (\boldsymbol{J}^{\otimes a_m})_{\alpha_m} - J_i \sum_{j,m} \sum_{\alpha_m} J_j^{c_m+1} m\boldsymbol{\mu}_{j,\alpha_m} (\boldsymbol{J}^{\otimes a_m})_{\alpha_m} \tag{10}$$

where $m\boldsymbol{\mu}_{i,\alpha_m} = A_m \langle \mathrm{x}_i^{b_m} (\mathbf{x}^{\otimes a_m})_{\alpha_m} \rangle_{\mathbf{x}}$. At steady states where $J_i \neq 0$,

$$\sum_m J_i^{c_m} \sum_{\alpha_m} m\mu_{i,\alpha_m} (\boldsymbol{J}^{\otimes a_m})_{\alpha_m} = \lambda J_i, \text{ where } \lambda(\boldsymbol{J}, \{m\boldsymbol{\mu}\}) = \sum_m \sum_{j,\alpha_m} J_j^{c_m+1} m\boldsymbol{\mu}_{j,\alpha_m} (\boldsymbol{J}^{\otimes a_m})_\alpha.$$

$$\tag{11}$$

If each $c_m = 0$, this is a kind of generalized tensor eigenequation:

$$\sum_{m,\alpha_m} m\boldsymbol{\mu}_{i,\alpha} (\boldsymbol{J}^{\otimes a_m})_\alpha = \lambda J_i \tag{12}$$

so that $\boldsymbol{J}$ is invariant under the combined action of the multilinear maps $_m\boldsymbol{\mu}$ (which are potentially of different orders). If $a_1 = a_2 = \cdots = a_m$, then this can be simplified to a tensor eigenvector equation by summing the input correlations $_m\boldsymbol{\mu}$. If different terms of the expansion of $f$ generate different-order input correlations, however, the steady states are no longer necessarily equivalent to tensor eigenvectors. If there exists a synaptic weight vector $\boldsymbol{J}$ that is an eigenvector of each of those input correlation tensors, $\sum_\alpha m\mu_{i,\alpha_m}(\boldsymbol{J}^{\otimes a_m})_{\alpha_m} = \lambda J_i$ for each $m$, then that configuration $\boldsymbol{J}$ is a steady state of the plasticity dynamics with each $c_m = 0$.

We next investigated whether these steady states were attractors in simulations of a learning rule with a contribution from two-point and three-point correlations ($\boldsymbol{a} = (1, 2)$, $\boldsymbol{b} = \boldsymbol{1}$, $\boldsymbol{c} = \boldsymbol{0}$, $\boldsymbol{A} = (1, 1/2)$ in eq. 9). As before, we used whitened natural image patches for the inputs $\mathbf{x}$ (fig. 1a). The two- and three-point correlations of those image patches had similar first eigenvalues, but the spectrum of the two-point correlation decreased more quickly than the three-point correlation (fig. 3a). The first three eigenvectors of the different correlations overlapped strongly (fig. 3b). This was not due to a trivial constant offset since the inputs were whitened. With this parameter set, the synaptic weights usually converged to the (shared) first eigenvector of the input correlations (fig. 3c,d, e (blue)).

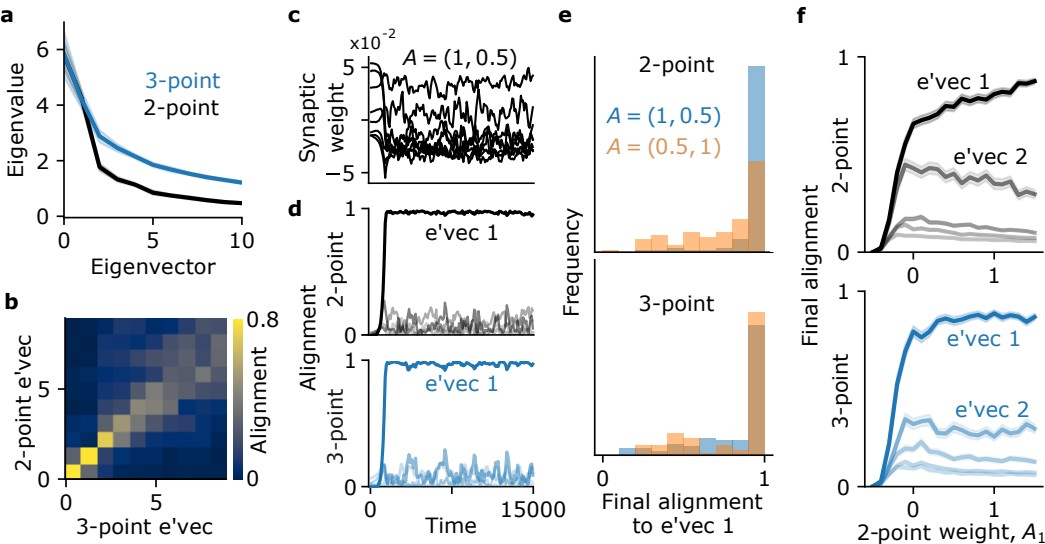

Figure 3: Multi-term Hebbian plasticity rule converges to input correlation tensors' shared eigenvectors. We study a learning rule with two terms, driven by the second- and third-order input correlations ($\boldsymbol{a} = (1, 2)$, $\boldsymbol{b} = \boldsymbol{1}$, $\boldsymbol{c} = \boldsymbol{0}$). **a)** Tucker spectra of the natural image patches' correlation tensors. Solid lines: mean over 10 samples of 200 image patches. Shaded regions: standard deviation. **b)** Overlap of the eigenvectors of the image patches' two- and three-point correlations (mean over 10 samples of 200 image patches). **c, d)** Example dynamics for the two-term learning rule. Transparency increases with the eigenvector number in **d**. **e)** Distribution of the final alignment to the first eigenvectors over 10 random initial conditions for each of 10 samples of image patches. **f)** Final alignment to the two- and three-point correlations' eigenvectors as a function of the weight on the two-point correlation, $A_1$. $A_2 = 1$.

We next asked how the weights of the different input correlations in the learning rule (the parameters $A_1, A_2$) affected the plasticity dynamics. When the learning rule weighted the inputs' three-point correlation more strongly than the two-point correlation ($\boldsymbol{A} = (1/2, 1)$), the dynamics converged almost always to the first eigenvector of the three-point correlation (fig. 3e, blue vs orange). Without loss of generality, we then fixed $A_2 = 1$ and varied the amplitude of $A_1$. As $A_1$ increased, the learning dynamics converged to equilibria increasingly aligned with the top eigenvectors of the input correlations (fig. 3f). For sufficiently negative $A_1$, the dynamics converged to steady states that were neither eigenvectors of the two-point input correlation nor any of the top 20 eigenvectors of the three-point input correlation (fig. 3f).

Earlier, we saw that in single-term learning rules, the only attractors were robust eigenvectors of the input correlation (theorem 1). The dynamics of eq. 2 usually converged to the first eigenvector

because it had the largest basin of attraction (corollaries 1.1, 1.2). Here, we saw that at least for some parameter sets, the dynamics of a multi-term generalized Hebbian rule may not converge to an input eigenvector. This suggested the existence of other attractors for the dynamics of eq. 10.

We next investigated the steady states of multi-term nonlinear Hebbian rules analytically. We focused on the case when the different input correlations generated by the learning rule all have a shared set of eigenvectors. In this case, those shared eigenvectors are all stable equilibria of eq. 10. They are not, however, the only stable equilibria. We see in a simple example that there can be equilibria that are linear combinations of those eigenvectors with all negative weights (in fig. 2c, the fixed point in the lower left quadrant on the unit circle). In fact, any stable equilibrium that is not a shared eigenvector must be such a negative combination. Our results are summarized in the following theorem:

**Theorem 2.** *In eq. 10 take $\boldsymbol{b} = 1, \boldsymbol{c} = 0, \boldsymbol{a} \in \mathbb{Z}_+^N$, and consider $N$ cubical, symmetric tensors, $_m\boldsymbol{\mu}$, each of order $a_m + 1$ for $m \in [N]$, that are mutually odeco into $R$ components:*

$$_m\boldsymbol{\mu} = \sum_{r=1}^{R} \lambda_{mr} \boldsymbol{U}_r \otimes \boldsymbol{U}_r \otimes \cdots \otimes \boldsymbol{U}_r \tag{13}$$

*with $\boldsymbol{U}^T\boldsymbol{U} = \boldsymbol{I}$. Let $\lambda_{mr} \geq 0$ and $\sum_m \lambda_{mr} > 0$ for each $m, r \in [N] \times [R]$. Let*

$$S(\boldsymbol{J}) = \sum_{i=1}^{R} (\boldsymbol{U}_i^T \boldsymbol{J})^2 \text{ and } L(\boldsymbol{J}) = \sum_{m=1}^{N} \sum_{i=1}^{R_m} \lambda_{mi} (\boldsymbol{U}_i^T \boldsymbol{J})^{a_m+1}. \tag{14}$$

*Then:*

1. *$\mathcal{S}^* = \{\boldsymbol{J} : S(\boldsymbol{J}) = 1 \wedge L(\boldsymbol{J}) > 0\}$ is an attracting set for eq. 10 and its basin of attraction includes $\{\boldsymbol{J} : L(\boldsymbol{J}) > 0\}$.*

2. *For each $k \in [R]$, $\boldsymbol{J} = \boldsymbol{U}_k$ is a stable equilibrium of eq. 10.*

3. *For each $k \in [R]$, $\boldsymbol{J} = -\boldsymbol{U}_k$ is a stable equilibrium of eq. 10 if $\sum_m {_m\lambda_k}(-1)^{a_m} < 0$ (and unstable if $\sum_m {_m\lambda_k}(-1)^{a_m} < 0$).*

4. *Any other stable equilibrium must have $\boldsymbol{U}_k^T\boldsymbol{J} \leq 0$ for each $k \in [R]$.*

The claims of theorem 2 are proven in appendix A.2. Similar to theorem 1, we see that the robust eigenvectors of each input correlation generated by the learning rule are stable equilibria of the learning dynamics. The complexity of eq. 10 has kept us from determining their basins of attraction. We can, however, make several guarantees. First, in a large region, the unit sphere is an attracting set for the dynamics of eq. 10. Second, the only stable fixed points are either the eigenvectors of $\pm\boldsymbol{\mu}$ or combinations of the eigenvectors of $\boldsymbol{\mu}$ with only nonpositive weights. This is in contrast to the situation where the learning rule has only one term; then theorem 1 guarantees that the only attractors are eigenvectors.

## 3 Discussion

We have analyzed biologically motivated plasticity dynamics that generalize the Oja rule. One class of these compute tensor eigenvectors. We proved that without a multiplicative weight-dependence in the plasticity, those eigenvectors are attractors of the dynamics (theorem 1, figs. 1, 2a, b). Contrary to Oja's rule, the first eigenvector of higher-order input correlations is not a unique attractor. Rather, each eigenvector $k$ has a finite basin of attraction, the size of which is proportional to $\lambda_k^{R/a-1}$. If there are $d$ codominant eigenvectors ($\lambda_1 = \lambda_2 = \ldots = \lambda_d$), the $d$-sphere they span is an attracting equilibrium manifold (corollary 1.4 in appendix A.2). Furthermore, steady states of any plasticity model with a finite Taylor polynomial in the neural output and inputs are generalized eigenvectors of multiple input correlations. These steady states are stable and attracting (theorem 2, figs. 2c, 3).

### 3.1 Spiking neurons and weight-dependence

While biological synaptic plasticity is certainly more complex than the simple generalized Hebbian rule of eq. 1, neural activity is also more complex than the linear model $n = \boldsymbol{J}^T\mathbf{x}$. We examined the

simple linear-nonlinear-Poisson spiking model and a generalized spike timing–dependent plasticity (STDP) rule ([45]; appendix A.3). Similar to eqs. 2 and 10, we can write the dynamical equation for $\boldsymbol{J}$ as a function of joint cumulant tensors of the input (eq. 56 in appendix A.3). These dynamics have a different structure than eqs. 2 and 10.

We focused here mainly on learning rules with no direct dependence on the synaptic weight ($c = 0$ in eq. 1, $\boldsymbol{c} = 0$ in eq. 9). When $c \neq 0$, the learning dynamics cannot be simply analyzed in terms of the loading onto the input correlations' eigenvectors. We studied the learning dynamics with weight-dependence for two simple families of input correlations: diagonal $\boldsymbol{\mu}$ and piecewise-constant rank one $\boldsymbol{\mu}$ (appendix A.4). In both cases, we found that with eigenvectors of those simple input correlations were also attractors of the plasticity rule. With diagonal input correlations, sparse steady states with one nonzero synapse are always stable and attracting when $a + c > 0$, but if $a + c \leq 0$ synaptic weights converge to solutions where all weights have the same magnitude (fig. A.4.1). With rank one input correlations, multiplicative weight-dependence can interfere with synaptic scaling and lead to an instability for the neurons' total synaptic amplitude (fig. A.4.2).

## 3.2 Related work and applications

There is a rich literature on generalized or nonlinear forms of Hebbian learning. We briefly discuss the most closely related results, to our knowledge. The family of Bienenstock, Cooper, and Munro (BCM) learning rules supplement the classic Hebbian model with a stabilizing sliding threshold for potentiation rather than synaptic scaling [59]. BCM rules balance terms driven by third- and fourth-order joint moments of the pre- and postsynaptic activity [60]. A triplet STDP model with rate-dependent depression and uncorrelated Poisson spiking has BCM dynamics [45] and can develop selective (sparse) connectivity in response to rate- or correlation-based input patterns [61]. If the input is drawn from a mixture model then under a BCM rule, the synaptic weights are guaranteed to converge to the class means of the mixture [62].

Learning rules with suitable postsynaptic nonlinearities can allow a neuron to perform independent component analysis (ICA) [63, 64]. These learning rules optimize the kurtosis of the neural response. In contrast, we show that a simple nonlinear Hebbian model learns tensor eigenvectors of higher-order input correlations. Those higher-order input correlations can determine which features are learned by gradient-based ICA algorithms [65]. Taylor and Coombes showed that a generalization of the Oja rule to higher-order neurons can also learn higher-order correlations [66], which can allow learning independent components [67]. In that model, however, the synaptic weights $\boldsymbol{J}$ are a higher-order tensor.

Computing the robust eigenvectors of an odeco tensor $\boldsymbol{\mu}$ by power iteration has $\mathcal{O}(K^{a+1})$ space complexity: it requires first computing $\boldsymbol{\mu}$. The discrete-time dynamics of eq. 1 correspond to streaming power iteration, with $\mathcal{O}(K)$ space complexity [68–70]). Eq. 2 defines limiting continuous-time dynamics for tensor power iteration, exposing the basins of attraction.

Oja's rule inspired a generation of neural algorithms for PCA and subspace learning [12, 13]. Local learning rules for approximating higher-order correlation tensors may also prove useful, for example in neuromorphic devices [71–73].

## Code availability

The code associated with figures one and three is available at `https://github.com/gocker/TensorHebb`.

## Acknowledgments

We thank the Allen Institute founder, Paul G. Allen, for his vision, encouragement and support.

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
