# A Appendix

## A.1 A nonlinear Hebbian learning rule: model and notation

We take a neuron receiving $K$ time-varying inputs $x_i(t)$, each filtered through a connection with synaptic weight $J_i$ to produce activity $n(t)$. We consider learning rules where the update of $J_i$ can depend on the postsynaptic activity $n(t)$, the local input $x_i(t)$, and the current synaptic weight $J_i(t)$. We encode these dependencies in a learning rule $f$:

$$f\left(n(t), x_i(t), J_i(t)\right) = n^a x_i^b J_i^c \tag{1}$$

where $a, b \in \mathbb{Z}^+$ and $c \in \mathbb{R}$. Finally, we assume that the synaptic weights are homeostatically regulated [9]. Together,

$$J_i(t + dt) = \frac{J_i(t) + (dt/\tau) \; f(n(t), x_i(t), J_i(t))}{\|\boldsymbol{J} + (dt/\tau) \, \boldsymbol{f}\|_p} \tag{2}$$

where $dJ_i = J_i(t + dt) - J_i(t)$, $\tau$ is a learning timescale, and $\|\mathbf{x}\|_p$ is the $\ell^p$ norm of $\mathbf{x}$.

We will model the neuron as a simple linear unit, $n(t) = (\boldsymbol{J}\mathbf{x})(t)$. Note that taking $n(t) = (\boldsymbol{J}\mathbf{x})^a(t)$ and $f\left(n(t), x_i(t), J_i(t)\right) = n(t)x_i^b(t)J_i^c(t)$ yields the same weight update as (eq. 1). As observed by [74], it is the composition of the neural nonlinearity and the output-dependent nonlinearity in the learning rule that determines the effective nonlinearity of the output-dependence in learning. A power-law neural transfer function has been shown to approximate biological models near their spiking threshold [75, 76].

We follow Oja [10] and expand in powers of $dt$ (equivalently, in $1/\tau$ or $dt/\tau$) which yields to linear order,

$$\tau \frac{dJ_i}{dt} = f\left(n(t), x_i(t), J_i(t)\right) - J_i \sum_j J_j |J_j|^{p-2} f\left(n(t), x_j(t), J_j(t)\right)$$

$$= n^a x_i^b J_i^c - J_i n^a \sum_j J_j |J_j|^{p-2} x_j^b J_j^c \tag{3}$$

We suppressed the time-dependence of $\mathbf{x}$ and $\boldsymbol{J}$ here and onwards. We will assume, as is standard, that learning is slow ($dt/\tau \ll 1$). In this case, individual changes in synaptic weights are small. If the inputs $\mathbf{x}$ are stationary (at least within a timescale $T$, $dt \ll T \ll \tau$) and have finite joint moments up to order $a + 1$, the dynamics average over the statistics of $\mathbf{x}$ [77] so that

$$\tau \dot{J}_i = J_i^c \sum_\alpha \mu_{i,\alpha} (\boldsymbol{J}^{\otimes a})_\alpha - J_i \sum_{j,\alpha} J_j^{c+1} |J_j|^{p-2} \mu_{j,\alpha} (\boldsymbol{J}^{\otimes a})_\alpha \tag{4}$$

where $\dot{J}_i(t) = (1/T) \int_t^{t+T} dt' \; (dJ_i/dt)(t')$ and $\mu_{i,\alpha} = \langle x_i^b (\mathbf{x}^{\otimes a})_\alpha \rangle_{\mathbf{x}} \approx (1/T) \int_t^{t+T} dt' \; x_i^b(t)(\mathbf{x}^{\otimes a})_\alpha(t)$ is a $(a + b)$-order joint moment of $\mathbf{x}$ and an $(a + 1)$-order tensor. The order of the tensor refers to its number of indices, so a vector is a first-order tensor and a matrix a second-order tensor. Since $\boldsymbol{\mu}$ is a correlation tensor of $\mathbf{x}$ each of its modes has the same range, $1, \ldots, K$. We also use multi-index notation: $\alpha = k_1, k_2, \ldots, k_a$. Sums over any index run from 1 to $K$ unless otherwise specified.

## A.2 Proofs

**Theorem 1.** *In eq. 2, take $(b, c) = (1, 0)$. Let $\boldsymbol{\mu}$ be a cubical, symmetric tensor of order $a + 1$ and orthogonally decomposable (odeco) into $R$ components:*

$$\boldsymbol{\mu} = \sum_{r=1}^R \lambda_r \left(\boldsymbol{U}_r\right)^{\otimes a+1} \tag{5}$$

*where $\boldsymbol{U}$ is a matrix of unit-norm orthogonal E-eigenvectors: $\boldsymbol{U}^T \boldsymbol{U} = \boldsymbol{I}$. Let $\lambda_i > 0$ for each $i \in [R]$ and $\lambda_i \neq \lambda_j \; \forall \; (i, j) \in [R] \times [R]$ with $i \neq j$. Then for each $k \in [R]$:*

1. *With any odd $a > 1$, $\boldsymbol{J} = \pm \boldsymbol{U}_k$ are attracting fixed points of eq. 2 and their basin of attraction is $\bigcap_{i \in [R] \backslash k} \left\{ \boldsymbol{J} : \left| \boldsymbol{U}_i^T \boldsymbol{J} / \boldsymbol{U}_k^T \boldsymbol{J} \right| < \left(\lambda_k / \lambda_i\right)^{1/(a-1)} \right\}$. Within that region, the separatrix of $+\boldsymbol{U}_k$ and $-\boldsymbol{U}_k$ is the hyperplane orthogonal to $\boldsymbol{U}_k^T$: $\{\boldsymbol{J} : \boldsymbol{U}_k^T \boldsymbol{J} = 0\}$.*

2. *With any even positive $a$, $\boldsymbol{J} = \boldsymbol{U}_k$ is an attracting fixed point of eq. 2 and its basin of attraction is $\left\{\boldsymbol{J} : \boldsymbol{U}_k^T \boldsymbol{J} > 0\right\} \bigcap_{i \in [R] \setminus k} \left\{\boldsymbol{J} : \boldsymbol{U}_i^T \boldsymbol{J} / \boldsymbol{U}_k^T \boldsymbol{J} < \left(\lambda_k/\lambda_i\right)^{1/(a-1)}\right\}$.*

3. *With any even positive $a$, $\boldsymbol{J} = \boldsymbol{0}$ is a neutrally stable fixed point of eq. 2 with basin of attraction $\left\{\boldsymbol{J} : \sum_{j=1}^{R}(\boldsymbol{U}_j^T \boldsymbol{J})^2 < 1 \wedge \boldsymbol{U}_k^T \boldsymbol{J} < 0 \; \forall \; k \in [R]\right\}$.*

*Proof.* Note that because the eigenvalues of $\boldsymbol{\mu}$ are distinct, $\boldsymbol{U}$ is unique [47]. Reshaping $\boldsymbol{\mu}$ from an order $a+1$ tensor with each fiber of length $K$ to $\boldsymbol{\mu}_{(n)}$, a $K \times K^a$ matrix with the rows equal to mode $n$ of $\boldsymbol{\mu}$, yields the matricized form [23]

$$\boldsymbol{\mu}_{(n)} = \boldsymbol{U}\boldsymbol{\Lambda}\left(\boldsymbol{U}^{\odot a}\right)^T \tag{6}$$

where $\boldsymbol{\Lambda} = \operatorname{diag}(\lambda_1, \ldots, \lambda_R)$ and $\odot$ is the columnwise Khatri-Rao product; $\boldsymbol{U}^{\odot a}$ is the $a$-fold Khatri-Rao product of $\boldsymbol{U}$ with itself, a $K^a \times R$ matrix. Let $\boldsymbol{J}(t) \equiv \boldsymbol{U}\boldsymbol{v}(t)$. From the mixed-product property of the Kronecker and dot products,

$$\boldsymbol{J}^{\otimes a} = \boldsymbol{U}^{\otimes a}\boldsymbol{v}^{\otimes a} \tag{7}$$

where $\otimes$ is the Kronecker product; $\boldsymbol{U}^{\otimes a}$ is the $a$-fold Kronecker product of $\boldsymbol{U}$ with itself, a $K^a \times R^a$ matrix. (For vectors, the Kronecker product is the vector outer product.) We insert the decomposition of $\boldsymbol{\mu}_{(n)}$ and this projection of $\boldsymbol{J}$ into the plasticity dynamics, eq. 2 with $c = 0$:

$$\begin{aligned}
\tau\dot{\boldsymbol{J}} &= \boldsymbol{\mu}_{(n)}\boldsymbol{J}^{\otimes a} - \boldsymbol{J} \circ \boldsymbol{J}^T \boldsymbol{\mu}_{(n)}\boldsymbol{J}^{\otimes a} \\
\tau\boldsymbol{U}\dot{\boldsymbol{v}} &= \boldsymbol{U}\boldsymbol{\Lambda}(\boldsymbol{U}^{\odot a})^T\boldsymbol{U}^{\otimes a}\boldsymbol{v}^{\otimes a} - \boldsymbol{U}\boldsymbol{v} \circ \boldsymbol{v}^T\boldsymbol{U}^T\boldsymbol{U}\boldsymbol{\Lambda}(\boldsymbol{U}^{\odot a})^T\boldsymbol{U}^{\otimes a}\boldsymbol{v}^{\otimes a}
\end{aligned} \tag{8}$$

where $\circ$ is the elementwise product. Since $\boldsymbol{U}$ is orthogonal so are $\boldsymbol{U}^{\otimes a}$ and $\boldsymbol{U}^{\odot a}$ and $(\boldsymbol{U}^{\odot a})^T\boldsymbol{U}^{\otimes a} = \boldsymbol{I}$, where $\boldsymbol{I}$ is a $R \times R^a$ identity matrix that picks out diagonal elements of $\boldsymbol{v}^{\otimes a}$. Let $\boldsymbol{\Sigma} = \boldsymbol{\Lambda}\boldsymbol{I}$, so

$$\begin{aligned}
\tau\boldsymbol{U}\dot{\boldsymbol{v}} &= \boldsymbol{U}\boldsymbol{\Lambda}\boldsymbol{I}\boldsymbol{v}^{\otimes a} - \boldsymbol{U}\boldsymbol{v} \circ \boldsymbol{v}^T\boldsymbol{\Lambda}\boldsymbol{I}\boldsymbol{v}^{\otimes a} \\
\tau\dot{\boldsymbol{v}} &= \boldsymbol{\Sigma}\boldsymbol{v}^{\otimes a} - \boldsymbol{v} \circ \boldsymbol{v}^T\boldsymbol{\Sigma}\boldsymbol{v}^{\otimes a}, \text{ or} \\
\tau\dot{v}_i &= v_i^a\lambda_i - v_i\sum_{j=1}^{R}\lambda_j v_j^{a+1}
\end{aligned} \tag{9}$$

Note that for any $i$, $v_i = 0$ is an equilibrium. In the standard Oja rule ($a = 1$), all other equilibria are on the unit sphere, $\mathcal{S} = \{\boldsymbol{v} : \sum_i v_i^2 = 1\}$, which is a globally attracting manifold [11]. Is this still the case? Let

$$S(\boldsymbol{v}) = \sum_i v_i^2, \; L(\boldsymbol{v}) = \sum_j \lambda_j v_j^{a+1} \tag{10}$$

so

$$\frac{\tau}{2}\dot{S} = L\left(1 - S\right) \tag{11}$$

If $a$ is odd, $L(\boldsymbol{v}) > 0$ for any $\boldsymbol{v}$, so $S = 1$ is a global attractor for $S$ and $\mathcal{S}$ is a globally attracting manifold for $\boldsymbol{v}$. If $a$ is even, $\mathcal{S}$ is attracting from regions where $L(\boldsymbol{v}) > 0$ but repelling from regions where $L(\boldsymbol{v}) < 0$.

What points in $\mathcal{S}$ are fixed points? From eq. 9, we have for each $i$ either $v_i^* = 0$ or it obeys the fixed point equation

$$v_i^* = \left(L^*/\lambda_i\right)^{1/(a-1)} \tag{12}$$

where $L^* = L(\boldsymbol{v}^*)$. This implies that if $v_i^* \neq 0$,

$$v_i^* = \pm\lambda_i^{1/(1-a)}\left(\sum_{j:v_j^* \neq 0}\lambda_j^{2/(1-a)}\right)^{-1/2} \tag{13}$$

In what follows, we will see that the sparse points on $\mathcal{S}$, with one $v_k = 1$ ($\pm 1$ if $a$ odd) and the rest at 0, are the only attractors and determine their basins of attraction.

Are the sparse points stable? The Jacobian of eq. 9 is

$$\frac{\partial \dot{v}_i}{\partial v_k} = \delta_{ik}\left(a\lambda_i v_i^{a-1} - (a+2)\lambda_i v_i^{a+1} - \sum_{j \neq i}\lambda_j v_j^{a+1}\right) + (1 - \delta_{ik})v_i(a+1)\lambda_k v_k^a \tag{14}$$

At a sparse fixed point, the Jacobian is diagonal. At a sparse $\boldsymbol{v}^*$ with 1 at element $j$, the Jacobian eigenvalues are $-2\lambda_j$ and, with multiplicity $R - 1$, $-\lambda_j$ so those positive sparse points are stable. At a sparse vector with -1 at element $j$, the Jacobian eigenvalues are $2\lambda_j(-1)^a$, and with multiplicity $R - 1$, $\lambda_j(-1)^a$. So the negative sparse points are stable if $a$ is odd and unstable if $a$ is even.

What are the basins of attraction of the stable sparse points? Can any non-sparse equilibria be stable? Following [11], we consider the change of variables

$$y_i = \frac{v_i}{v_k}, \ i \neq k \tag{15}$$

for some $v_k \neq 0$. We examine the joint dynamics of the loading onto the normalizing eigenvector, $v_k$, and the relative loadings onto the other eigenvectors, $y_i$. These are

$$\tau \dot{y}_i = v_k^{a-1} y_i \left(y_i^{a-1}\lambda_i - \lambda_k\right), \ i \neq k$$

$$\tau \dot{v}_k = v_k^a \left(\lambda_k - v_k^2\left(\lambda_k + \sum_{j \neq k}\lambda_j y_j^{a+1}\right)\right) \tag{16}$$

The nullclines for $v_k$ are $v_k^* \in \left\{0, \pm\sqrt{\lambda_k/(\lambda_k + \sum_{j \neq k}\lambda_j y_j^{a+1})}\right\}$ ($v_k = 0$ is a hyperplane equilibrium for the whole system, but the coordinate transform is singular at it). Checking $\text{sign}(\dot{v}_k)$ on either side of $v_k^* = \pm\sqrt{\lambda_k/(\lambda_k + \sum_{j \neq k}\lambda_j y_j^{a+1})}$ reveals that those nullclines for $v_k$ are both attracting with respect to $v_k$ for any finite $y_i$.

The stability of $v_k^* = 0$ depends on the parity of $a$. The $\boldsymbol{y}$-nullclines are $y_i^* \in \left\{0, \pm(\lambda_k/\lambda_i)^{1/(a-1)}\right\}$. (If $a$ is odd we have both roots, while if $a$ is even we have only the + root.) These nullclines also depend on the parity of $a$.

Case 1: odd $a$. First consider $v_k^* = 0$. Note that with $a$ odd, $\lambda_k + \sum_{j=2}^{R}\lambda_j y_j^{a+1} \geq \lambda_k > 0$. From eq. 16, $\text{sign}(\dot{v}_k) = \text{sign}(v_k)$ when $a$ is odd. So, $v_k$ is repelled from 0.

Now consider the $\boldsymbol{y}$-nullclines. For any $v_k \neq 0$, checking the sign of $\dot{y}_i$ reveals that $y_i = 0$ is an attractor and $y_i = \pm(\lambda_k/\lambda_i)^{1/(a-1)}$ are both repellers. Recall that $v_k^* = \pm\sqrt{\lambda_k/(\lambda_k + \sum_{j \neq k}\lambda_j y_j^{a+1})}$ is attracting along $v_k$ for any finite $\boldsymbol{y}$. Together, the only attracting equilibria in $v_k, \boldsymbol{y}$ can be at $v_k = \pm 1$ and each $y_i = 0$. Since the columns of $\boldsymbol{U}$ are orthonormal, $v_k = \boldsymbol{U}_k^T \boldsymbol{J} = \pm 1$ implies that $\boldsymbol{J} = \pm \boldsymbol{U}_k$. The basin of attraction of these equilibria are defined by the other, unstable nullclines.

Back in the space of $\boldsymbol{v}$, those are the unstable hyperplanes $v_i/v_k = \pm(\lambda_k/\lambda_i)^{1/(a-1)}$ and the repelling nullcline $v_k^* = 0$. Each unique pair $v_i, v_k$ generates one such pair of unstable hyperplanes, all passing through the origin. All of the equilibria points $\boldsymbol{v}^* \in \mathcal{S}$ identified earlier, with at least two nonzero elements, lie on at least one of those unstable hyperplanes and are thus unstable. Since $\mathcal{S}$ is a global attractor, these $R(R - 1) + 1$ hyperplanes partition $\mathbb{R}^R$ into the basins of attraction of each sparse point with one $v_k = \pm 1$ and the others 0, which are the only attractors.

Case 2: even $a$. Let $v_k > 0$ in the definition of $\boldsymbol{y}$. (If $L(\boldsymbol{v}) > 0$ and $a$ even, at least one $v_k$ must be positive.) Checking the sign of $\dot{y}_i$ then reveals that for each $i \neq k$, $y_i^* = 0$ is an attractor while $y_i^* = (\lambda_k/\lambda_i)^{1/(a-1)}$ is repelling. So the hyperplane $v_i = v_k(\lambda_k/\lambda_i)^{1/(a-1)}$ is unstable. Each unique pair of axes $v_i, v_k$ has such an unstable hyperplane where $v_k > 0$.

If $v_k(0) > 0$, $v_k$ cannot cross zero since $v_k = 0$ is an equilibrium for the whole system. Furthermore, if $v_k = \epsilon w_k$,

$$\tau \dot{w}_k = \epsilon^{a-1}\lambda_k w_k^a + \mathcal{O}(\epsilon^{a+1}) \tag{17}$$

so $v_k$ is repelled by 0 from above. So if $v_k > 0$ and each $y_i < (\lambda_k/\lambda_i)^{1/(a-1)}$, the relative loadings $y_i$ will all approach zero. Let each $y_i = 0$, so

$$\tau \dot{v}_k = \lambda_k v_k^a \left(1 - v_k^2\right) \tag{18}$$

with a stable equilibrium at $v_k = 1$. (The equilibrium $v_k = -1$ is excluded by construction.) Each sparse point with $v_k = 1$, $\boldsymbol{y} = \boldsymbol{0}$ corresponds to a sparse equilibrium for the loadings with $v_i = 0$, $i \neq k$. Together, the only attracting equilibria in $v_k, \boldsymbol{y}$ can be at $v_k = \pm 1$ and each $y_i = 0$. Since the columns of $\boldsymbol{U}$ are orthonormal, $v_k = \boldsymbol{U}_k^T \boldsymbol{J} = \pm 1$ implies that $\boldsymbol{J} = \pm \boldsymbol{U}_k$. The basin of attraction of these equilibria are defined by the other, unstable nullclines.

Each positive sparse point, with $v_k = 1$, lies inside a section of $\mathbb{R}^R$ bounded by the $R - 1$ repelling hyperplanes $v_i = v_k (\lambda_k/\lambda_i)^{1/(a-1)}$ and/or the repelling axis $v_k = 0$, and each such region contains one such sparse point. So, those hyperplanes divide $\mathbb{R}^R$ into the basins of attraction for the columns of $\boldsymbol{U}$.

Finally, $\boldsymbol{v} = \boldsymbol{0}$ is an equilibrium of eq. 9. At $\boldsymbol{v} = \boldsymbol{0}$, the Jacobian of *eq.* 9 is identically 0. Let $\boldsymbol{v} < 0$ elementwise. Then $L(\boldsymbol{v}) < 0$ so if $S(\boldsymbol{v}) < 1$ then $\dot{S} < 0$ (eq. 11). As $S(\boldsymbol{v}) \to 0$, no $v_k$ can become positive because $v_k = 0$ is an equilibrium. In the section under $S$ with $\boldsymbol{v} < 0$, $\boldsymbol{v}$ thus approaches the origin. $\qquad\square$

**Corollary 1.1.** *Let $V_k$ be the relative volume of the basin of attraction for $\boldsymbol{J}^* = \boldsymbol{U}_k^T$. For odd $a > 1$,*

$$V_k = R^{-1} \prod_{i=1}^{R} \left( \frac{\lambda_k}{\lambda_i} \right)^{1/(a-1)} \tag{19}$$

*Proof.* We will compute the volume $V_k$ of the basin of attraction for $\boldsymbol{U}_k^T$ directly. Take $a$ odd. Then from theorem 1,

$$V_k = \int D\boldsymbol{J} \prod_{\substack{i=1 \\ i \neq k}}^{R} \theta \left( C_{ik} - \left| \frac{\boldsymbol{U}_i^T \boldsymbol{J}}{\boldsymbol{U}_k^T \boldsymbol{J}} \right| \right) \tag{20}$$

where $D\boldsymbol{J} = \prod_{i=1}^{K} dJ_i$ and $C_{ik} = \left( \frac{\lambda_k}{\lambda_i} \right)^{1/(a-1)}$. We change variables to the relative loadings, $\boldsymbol{v} = \boldsymbol{U}^T \boldsymbol{J}$; the Jacobian factor is $\text{vol}(\boldsymbol{U})$, the product of the singular values of $\boldsymbol{U}$. Since $\boldsymbol{U}$ is orthogonal, its singular values are all 1. The integrals over $v_i, i \neq k$, all factorize:

$$\int_{-\infty}^{\infty} dv_i \, \theta \left( C_{ik}|v_k| - |v_i| \right) = 2C_{ik}|v_k| \tag{21}$$

which leaves

$$V_k = 2^{R-1} \left( \prod_{i \neq k} C_{ik} \right) \int_{-\infty}^{\infty} dv_k \, |v_k|^{R-1} \tag{22}$$

We choose bounds $-x, x$ for $\int dv_k$ and compute $V(x)$:

$$V_k(x) = R^{-1}(2x)^R \prod_{i \neq k} C_{ik} \tag{23}$$

Note that $C_{kk} = 1$, so $\prod_{i \neq k} C_{ik} = \prod_i C_{ik}$. $V_k(x)$ is the volume of the set of weight vectors with projection at least $x$ onto $\boldsymbol{U}_k^T$ that will converge to $\boldsymbol{U}^k$. We recognize the factor of $(2x)^R$ as the volume of an $R$-dimensional hypercube with edge lengths $2x$. Normalizing by the volume of the hypercube concludes the calculation.

For even $a$, integrating each $v_j$ over $(0, \infty)$ and normalizing by the hypercube volume $x^R$, rather than $(2x)^R$, yields the same result. The region with all $v_i > 0$ is certainly part of the basin of attraction of $v_k = 1$, but not the whole basin which requires only $v_k > 0$. This result thus provides a lower bound of the volume of the basins of attraction for even $a$. $\qquad\square$

**Corollary 1.2.** *Let $V_k$ be the relative volume of the basin of attraction for $\boldsymbol{J}^* = \boldsymbol{U}_k^T$. For even positive $a$,*

$$V_k = 2^{1-R} \left( R^{-1} \prod_i \left( \frac{\lambda_k}{\lambda_i} \right)^{1/(a-1)} + (R-1)^{-1} \sum_{j \neq k} \prod_{i \neq j} \left( \frac{\lambda_k}{\lambda_i} \right)^{1/(a-1)} \right.$$
$$\left. + (R-2)^{-1} \sum_{j,l \neq k} \prod_{i \neq j,l} \left( \frac{\lambda_k}{\lambda_i} \right)^{1/(a-1)} + \ldots + 1 \right) \tag{24}$$

*Proof.* For even $a$, we compute

$$V_k = \int DJ \, \theta \left( \boldsymbol{U}_k^T \boldsymbol{J} \right) \prod_{\substack{i=1 \\ i \neq k}}^{R} \theta \left( C_{ik} - \frac{\boldsymbol{U}_i^T \boldsymbol{J}}{\boldsymbol{U}_k^T \boldsymbol{J}} \right) \tag{25}$$

as in corollary 1.1. The integrals over $v_i$, $i \neq k$ all factorize:

$$\int_{-\infty}^{\infty} dv_i \, \theta \left( C_{ik} v_k - v_i \right) \tag{26}$$

and we choose a bound of $-x$ for them, leaving

$$V_k(x) = \int_0^x dv_k \prod_{i \neq k} \left( C_{ik} v_k + x \right) \tag{27}$$

The proper normalization here is by $2^{R-1} x^R$, with a factor of $x$ from $\int dv_k$ and $(2x)^{R-1}$ from the integrals $\int dv_i$.

$\square$

**Corollary 1.3.** *Let $\mathcal{S}$ be the unit $R$-sphere in $\mathbb{R}^R$ and $\mathcal{S}_k$ be the section of $\mathcal{S}$ in the basin of attraction of $\boldsymbol{J}^* = \boldsymbol{U}_k^T$. If $a$ is odd, the surface area of $\mathcal{S}_k$ is*

$$A_k = 2 \tan^{-1} C_{jk} \prod_{\substack{i=1 \\ i \neq j}}^{R-1} \Gamma \left( \frac{R-i}{2} \right) \left( \frac{\sqrt{\pi}}{\Gamma \left( \frac{R-i+1}{2} \right)} - \frac{\left( 1 + C_{ik}^2 \right)^{j-R/2}}{\Gamma \left( \frac{R-i+2}{2} \right)} {}_2F_1 \left( \frac{1}{2}, \frac{R-i}{2}, \frac{R-j+2}{2}, \frac{1}{1+C_{ik}^2} \right) \right). \tag{28}$$

*If $a$ is even, the surface area of $\mathcal{S}_k$ is*

$$S_k = \left( \tan^{-1} C_{jk} + \frac{\pi}{2} \right) \prod_{\substack{i=1 \\ i \neq j}}^{R-1} \frac{1}{2} \Gamma \left( \frac{R-i}{2} \right) \left( \frac{2\sqrt{\pi}}{\Gamma \left( \frac{R-i+1}{2} \right)} - \frac{\left( 1 + C_{ik}^2 \right)^{j-R/2}}{\Gamma \left( \frac{R-i+2}{2} \right)} {}_2F_1 \left( \frac{1}{2}, \frac{R-i}{2}, \frac{R-j+2}{2}, \frac{1}{1+C_{ik}^2} \right) \right) \tag{29}$$

*Proof.* We calculate the surface area by transforming to spherical coordinates with an azimuthal angle $\theta_j \in [0, 2\pi)$ and $R - 2$ polar angles $\theta_i \in [0, \pi]$. Without loss of generality, we place $\boldsymbol{U}_k^T$ at $\theta_i = 0, \theta_j = \pi/2$ for each $j$.

Take $a$ odd. The unstable hyperplanes bounding the basin of attraction for $\boldsymbol{U}_k^T$ are defined by the azimuthal angle $\theta_i = \pm \tan^{-1} (\lambda_k / \lambda_i)^{1/a-1}$ and polar angles $\theta_i = \pi/2 \pm \tan^{-1} (\lambda_k / \lambda_i)^{1/a-1}$. Let $C_{ik} = (\lambda_k / \lambda_i)^{1/a-1}$. The surface area of $\mathcal{S}_k$ with $a$ odd is:

$$S_k = \int_{-\tan^{-1} C_{jk}}^{\tan^{-1} C_{jk}} d\theta_j \prod_{\substack{i=1 \\ i \neq j}}^{R-1} \int_{\frac{\pi}{2} - \tan^{-1} C_{ik}}^{\frac{\pi}{2} + \tan^{-1} C_{ik}} d\theta_i \, \sin^{R-i-1} \theta_i \tag{30}$$

Take $a$ even. The unstable hyperplanes bounding the basin of attraction for $\boldsymbol{U}_k^T$ are defined by the azimuthal angle $\theta_i = \tan^{-1} (\lambda_k / \lambda_i)^{1/a-1}$ and the polar angles $\theta_i = \pi/2 + \tan^{-1} (\lambda_k / \lambda_i)^{1/a-1}$. The other bounds for the basin of attraction are that they have positive loadings, $\boldsymbol{U}_k^T \boldsymbol{J} > 0$. Those correspond to the azimuthal angle $-\pi/2$ and the polar angles $\pi$. The surface area of $\mathcal{S}_k$ with $a$ even is:

$$S_k = \int_{-\frac{\pi}{2}}^{\tan^{-1} C_{jk}} d\theta_j \prod_{\substack{i=1 \\ i \neq j}}^{R-1} \int_{\frac{\pi}{2} - \tan^{-1} C_{ik}}^{\pi} d\theta_i \, \sin^{R-i-1} \theta_i \tag{31}$$

$\square$

**Remark.** *For small eigenvalues $\lambda_k$, the limits of integration for the polar factors (e.g., $\int_{\frac{\pi}{2}-\tan^{-1}C_{ik}}^{\frac{\pi}{2}+\tan^{-1}C_{ik}} d\theta_i \ \sin^{R-i-1}\theta_i$) approach 0 and $\pi$. (For even $a$, the upper limit is always $\pi$.) For small $\lambda_k$, those polar factors thus approach 1. This raises the hope that those products might be truncated. The number of eigenvalues is, however, exponentially large in K: $a^K-1/a-1$ [50, 51], and standard algorithms for computing the singular value decomposition of a tensor have space complexity $\mathcal{O}(K^{a+1})$. We computed the first 20 singular vectors (eigenvectors) of $\boldsymbol{\mu}$, and they did not decay to negligible values within those.*

**Corollary 1.4.** *In eq. 2, take $(b,c) = (1,0)$. Let $\boldsymbol{\mu}$ be a cubical, symmetric tensor of order $a+1$ and rank $R$, as in theorem 1, but with $d$ equal eigenvalues. Let $D$ be the index set of those equal eigenvalues. Let $\mathcal{S}_D \in \mathbb{R}^R$ be the unit $d$-sphere spanned by $\{\boldsymbol{U}_j^T : j \in D\}$ and let $-\mathcal{S}_D$ be the unit $d$-sphere spanned by $\{-\boldsymbol{U}_j^T : j \in D\}$. Then:*

1. *With any odd $a > 1$, $\mathcal{S}_D$ and $-\mathcal{S}_D$ are attracting equilibrium manifolds of eq. 2. The basin of attraction for $\mathcal{S}_D$ is $\bigcup_{k \in D} \bigcap_{i \notin D} \left\{ \boldsymbol{J} : - \left(\frac{\lambda_k}{\lambda_i}\right)^{1/(a-1)} < \frac{\boldsymbol{U}_i^T \boldsymbol{J}}{\boldsymbol{U}_k^T \boldsymbol{J}} < \left(\frac{\lambda_k}{\lambda_i}\right)^{1/(a-1)} \right\}$. The basin of attraction for $-\mathcal{S}_D$ is $\bigcup_{k \in D} \bigcap_{i \notin D} \left\{ \boldsymbol{J} : - \left(\frac{\lambda_k}{\lambda_i}\right)^{1/(a-1)} > \frac{\boldsymbol{U}_i^T \boldsymbol{J}}{\boldsymbol{U}_k^T \boldsymbol{J}} > \left(\frac{\lambda_k}{\lambda_i}\right)^{1/(a-1)} \right\}$.*

2. *With any even positive $a$, $\mathcal{S}_d$ is an attracting equilibrium manifold of eq. 2 and its basin of attraction is $\bigcup_{k \in D} \left( \{\boldsymbol{J} : \boldsymbol{U}_k^T \boldsymbol{J} > 0\} \bigcap_{i \notin D} \{\boldsymbol{J} : \frac{\boldsymbol{U}_i^T \boldsymbol{J}}{\boldsymbol{U}_k^T \boldsymbol{J}} < \left(\frac{\lambda_k}{\lambda_i}\right)^{1/(a-1)} \} \right)$.*

*Proof.* Since $d$ eigenvalues of $\boldsymbol{\mu}$ are equal, the eigendecomposition of $\boldsymbol{\mu}$ is not unique. Call $\boldsymbol{U}_D^T$ be the set of the $d$ eigenvectors with equal eigenvalues. Let

$$\boldsymbol{U}' = \boldsymbol{U}\boldsymbol{T} \tag{32}$$

where $\boldsymbol{T}$ is an orthogonal transformation within the subspace spanned by $\boldsymbol{U}_D^T$. For any such $\boldsymbol{T}$, the columns of $\boldsymbol{U}'$ are also eigenvectors of $\boldsymbol{\mu}$. Note that for any $i \notin D$, $\boldsymbol{U}_i'^T = \boldsymbol{U}_i^T$.

As in the proof of theorem 1, let $\boldsymbol{J} = \boldsymbol{U}'\boldsymbol{v}$. Pick one $k \in D$ and choose $\boldsymbol{T}$ such that $v_j = 0$ for each $j \in D$, $j \neq k$. This is a fixed point for the $d-1$ loadings $v_j$. For the remaining $R-d+1$ loadings, the proof of theorem 1 follows.

In particular, for odd $a$, $v_k = 1$ is an attractor with basin of attraction $\bigcap_{i \in [R] \setminus D} \left\{ \boldsymbol{J} : - (\lambda_k/\lambda_i)^{1/(a-1)} < v_i/v_k < (\lambda_k/\lambda_i)^{1/(a-1)} \right\}$. This holds for each $k \in D$. Together, the basin of attraction for $\boldsymbol{U}_D^T$ is the union of those basins of attraction. Similarly, the basin of attraction for $v_k = -1$ is $\bigcap_{i \in [R] \setminus D} \left\{ \boldsymbol{J} : - (\lambda_k/\lambda_i)^{1/(a-1)} > v_i/v_k > (\lambda_k/\lambda_i)^{1/(a-1)} \right\}$, and the basin of attraction for $-\boldsymbol{U}_D^T$ is the union of those.

Recall that the eigendecomposition of $\boldsymbol{\mu}$ is invariant under orthogonal transfomations [42]. That is, prior to choosing $\boldsymbol{T}$ above, the eigenvectors $\boldsymbol{U}_D^T$ can be replaced by any unit-norm linear combination thereof. Any point on $\mathcal{S}_D$ or $-\mathcal{S}_D$ is thus an attractor with the same basin of attraction defined above.

For even $a$, the same argument applies; the boundaries of the basins of attraction are as specified in theorem 1.

$\square$

**Theorem 2.** *In eq. 10, take $\boldsymbol{b} = 1, \boldsymbol{c} = 0, \boldsymbol{a} \in \mathbb{Z}_+^N$, and consider $N$ cubical, symmetric tensors, $_m\boldsymbol{\mu}$, each of order $a_m + 1$ for $m \in [N]$, that are mutually orthogonally decomposable into $R$ components:*

$$_m\boldsymbol{\mu} = \sum_{r=1}^{R} \lambda_{mr} \boldsymbol{U}_r^T \otimes \boldsymbol{U}_r^T \otimes \cdots \otimes \boldsymbol{U}_r^T \tag{33}$$

with $\|\boldsymbol{U}_r\|_2 = 1$ for each $r \in [R]$ and $\boldsymbol{U}_i^T \boldsymbol{U}_j = 0$ for $i \neq j$. Let $\lambda_{mr} \geq 0$ and $\sum_m \lambda_{mr} > 0$ for each $m, r \in [N] \times [R]$. Let

$$S(\boldsymbol{J}) = \sum_{i=1}^{R} (\boldsymbol{U}_i^T \boldsymbol{J})^2, \; L(\boldsymbol{J}) = \sum_{m=1}^{N} \sum_{i=1}^{R_m} \lambda_{mi} (\boldsymbol{U}_i^T \boldsymbol{J})^{a_m+1} \tag{34}$$

*Then:*

1. $\mathcal{S}^* = \{\boldsymbol{J} : S(\boldsymbol{J}) = 1 \wedge L(\boldsymbol{J}) > 0\}$ *is an attracting set for eq.* 10 *and its basin of attraction includes* $\{\boldsymbol{J} : L(\boldsymbol{J}) > 0\}$.

2. *For each* $k \in [R]$, $\boldsymbol{J} = \boldsymbol{U}_k$ *is a stable equilibrium of eq.* 10

3. *For each* $k \in [R]$, $\boldsymbol{J} = -\boldsymbol{U}_k$ *is a stable equilibrium of eq.* 10 *if* $\sum_m m \lambda_k (-1)^{a_m} < 0$ *(and unstable if* $\sum_m m \lambda_k (-1)^{a_m} < 0$)

4. *Any other stable equilibrium must have* $\boldsymbol{U}_k^T \boldsymbol{J} \leq 0$ *for each* $k \in [R]$.

*Proof.* We will prove the claims in the order of their statement in theorem 2. Let $\boldsymbol{J}(t) = \boldsymbol{U}\boldsymbol{v}(t)$; we again study the dynamics for the loadings:

$$\tau \dot{v}_i = \sum_m \lambda_{mi} v_i^{a_m} - v_i \sum_{m,j} \lambda_{mj} v_j^{a_m+1} \tag{35}$$

Let

$$S(\boldsymbol{v}) = \sum_i v_i^2, \; L(\boldsymbol{v}) = \sum_{m,j} \lambda_{mj} v_j^{a_m+1} \tag{36}$$

At a fixed point for $\boldsymbol{v}$, $S$ and $L$ must also be at a fixed point. The dynamics of $S$ are

$$\frac{\tau}{2} \dot{S} = L(1-S) \tag{37}$$

with fixed points at $S = 1, L = 0$. Let $\mathcal{S} = \{\boldsymbol{v} : S(\boldsymbol{v}) = 1\}$, the unit sphere, and $\mathcal{L} = \{\boldsymbol{v} : L(\boldsymbol{v}) = 0\}$. All fixed points $\boldsymbol{v}^*$ must be in $\mathcal{S}$ or $\mathcal{L}$. A fixed point has each $v_i$ at a root of $\sum_m \lambda_{mi} v_i^{a_m} - v_i(\boldsymbol{v})$. Furthermore, from eq. 35, we have that at a fixed point for any $i$, either $v_i = 0$ or it obeys the fixed point equation

$$L(\boldsymbol{v}) = \sum_m \lambda_{mi} v_i^{a_m-1} \tag{38}$$

$\mathcal{S}$ is attracting from above the boundary set $\mathcal{L} = \{\boldsymbol{v} : L(\boldsymbol{v}) = 0\}$. If $\boldsymbol{v}$ starts above $\mathcal{L}$, will it remain so? Is $\mathcal{L}$ attracting or repelling? Let $L(\boldsymbol{v}^*) = \epsilon$. Then

$$\tau \dot{v}_i = \sum_m \lambda_{mi} v_i^{a_m} + \mathcal{O}(\epsilon) \tag{39}$$

Let $\boldsymbol{v} = \boldsymbol{v}^* + \epsilon \boldsymbol{w}$, where $\boldsymbol{v}^* \in \mathcal{L}$ and $w_i = \sum_m \lambda_{mi} (v_i^*)^{a_m}$, so

$$L(\boldsymbol{v}^* + \epsilon \boldsymbol{w}) = \epsilon \sum_{m,n,j} \lambda_{mj}(v_j^*)^{a_m} \lambda_{nj}(v_j^*)^{a_n} + \mathcal{O}(\epsilon^2)$$

$$= \epsilon \sum_j \left( \sum_m \lambda_{mj}(v_j^*)^{a_m} \right)^2 + \mathcal{O}(\epsilon^2) \geq 0 \tag{40}$$

Points $\boldsymbol{v}^* \in \mathcal{L}$, if perturbed, will either 1) move above $\mathcal{L}$ or 2) if $\sum_j \left( \sum_m \lambda_{mj}(v_j^*)^{a_m} \right)^2 = 0$, stay on $\mathcal{L}$. So if $L(\boldsymbol{v}) > 0$ at some time $t$, $L(\boldsymbol{v}) \geq 0$ for all subsequent times and $\mathcal{S}^* = \{\boldsymbol{v} \in \mathcal{S} | L(\boldsymbol{v}) \geq 0\}$ is an attracting set for $\boldsymbol{v}$.

The sparse vectors $\boldsymbol{v}^*$ with one element at $\pm 1$ and the others at 0 are in $\mathcal{S}$. They correspond to equilibria for $\boldsymbol{J}$ at the columns of $\pm \boldsymbol{U}$. Are those equilibria stable? The Jacobian of eq. 35 is

$$\frac{\partial \dot{v}_i}{\partial v_k} = \delta_{ik} \sum_m \left( a_m \lambda_{mi} v_i^{a_m-1} - (a_m + 2)\lambda_{mi} v_i^{a_m+1} - \sum_{j \neq i} \lambda_{mj} v_j^{a_m+1} \right) + (1 - \delta_{ik}) v_i \sum_m (a_m + 1)\lambda_{mk} v_k^{a_m}$$

$$= \delta_{ik} \sum_m a_m \lambda_{mi} v_i^{a_m-1} \left(1 - v_i^2\right) + (1 - \delta_{ik}) v_i \left( L v_k + \sum_m a_m \lambda_{mk} v_k^{a_m} \right) \tag{41}$$

where we used the fixed point condition eq. 38. At a sparse fixed point, the Jacobian is diagonal. At a sparse $v^*$ with 1 at element $j$, the Jacobian eigenvalues are $-2\sum_m \lambda_{mj}$ and, with multiplicity $R-1$, $-\sum_m \lambda_{mj}$ so those are stable. At a sparse vector with -1 at element $j$, the Jacobian eigenvalues are $2\sum_m \lambda_{mj}(-1)^{a_m}$ and, with multiplicity $R-1$, $\sum_m \lambda_{mj}(-1)^{a_m}$. So the sparse points with -1 at element $j$ are stable if $\sum_m \lambda_{mj}(-1)^{a_m} < 0$ and unstable if $\sum_m \lambda_{mj}(-1)^{a_m} > 0$.

Next we will study non-sparse equilibria. We again study the dynamics of the relative loadings $y_i = v_i/v_k, i \neq k$, for some $v_k \neq 0$:

$$\tau \dot{y}_i = y_i \sum_{m=1}^{N} v_k^{a_m-1} \left( \lambda_{mi} y_i^{a_m-1} - \lambda_{mk} \right)$$

$$\tau \dot{v}_k = \sum_{m=1}^{N} v_k^{a_m} \left( \lambda_{mk} - v_k^2 \left( \lambda_{mk} + \sum_{j \neq k} \lambda_{mj} y_j^{a_m+1} \right) \right)$$

(42)

with nullclines for each $y_i$ at 0 and the other roots of $\sum_m v_k^{a_m-1} \left( \lambda_{mi} y_i^{a_m-1} - \lambda_{mk} \right)$, and nullclines for $v_k$ at 0 and the other roots of $\sum_m v_k^{a_m} \left( \lambda_{mk} - v_k^2 \left( \lambda_{mk} + \sum_{j \neq k} \lambda_{mj} y_j^{a_m+1} \right) \right)$. A fixed point for $v$ must also be a fixed point for $(v_k, y)$ for any $k$ with $v_k \neq 0$. If such a fixed point $\bar{v}$ has at least two nonzero elements $\bar{v}_i$, they must correspond to each $\bar{y}_i$ on a nonzero nullcline.

Consider the $y_i$-nullclines at the roots of $\sum_m v_k^{a_m-1} \left( \lambda_{mi} y_i^{a_m-1} - \lambda_{mk} \right)$. Let $y_i(t) = \bar{y}_i + \epsilon w_i(t)$, where $\sum_m v_k^{a_m-1} \left( \lambda_{mi} \bar{y}_i^{a_m-1} - \lambda_{mk} \right) = 0$. The dynamics of $w_i$ are

$$\tau \dot{w}_i = w_i \sum_m (a_m - 1) \lambda_{mi} v_k^{a_m-1} \bar{y}_i^{a_m-1} + \mathcal{O}(\epsilon)$$

(43)

These nullclines $\bar{y}_i$ are stable if $\sum_m (a_m - 1) \lambda_{mi} v_k^{a_m-1} \bar{y}_i^{a_m-1} < 0$, or equivalently $\sum_m (a_m - 1) \lambda_{mi} v_i^{a_m-1} < 0$. This is only possible if $v_i < 0$: *a condition directly on $v$*. A stable fixed point must thus have $v_i \leq 0$ for each $i \neq k$. This is true for any $k$ with $v_k \neq 0$. So, a stable fixed point must have $v_i \leq 0$ for each $i \in [R]$.

$\square$

## A.3  Spiking models

So far, we have discussed learning in a neuron model with two major simplifying assumptions. First, the neural output $n$ depended only on the current input $\mathbf{x}(t)$. Synaptic kinetics, however, exhibit nonzero time constants so that neural activity depends also on the recent history of its inputs. Second, the neural output was a continuous, linear function of the inputs. Cortical neurons, however, spike. We next relax these two assumptions. We introduce a generalized spike timing–dependent plasticity (STDP) rule:

$$f\left( n(t), \mathbf{x}_i(t), J_i(t) \right) = \mathbf{A}^T \left( n^a \mathbf{x}_i^b J_i^c \right)$$

(44)

where $\mathbf{A} = A(s)$ is the STDP kernel, a scalar function of each of the $a$ post-post lags, $b$ pre-post lags and $c$ synaptic weight lags. Here, the notation $\mathbf{A}^T \mathbf{X}$ denotes a functional inner product, integrating over the time lags of the STDP kernel $\mathbf{A}$ and the tensor $\mathbf{X}$ (eq. 47). We use this functional notation for simplicity and to emphasize the similarity with the simpler model of eq. 1. The case $a = 1, b = 1, c = 0$ corresponds to classic pair-based STDP [78–80] while $a = 2, b = 1, c = 0$ corresponds to triplet STDP [45]. The commonly used triplet STDP model has two terms: a pair-based depression and triplet-based potentiation. Here we first discuss STDP rules with one term and then consider an arbitrary expansion of a plasticity model in STDP kernels [44]. Similarly to for eq. 1, combining eq. 44 with a homeostatic normalization of the synaptic weights and a separation of timescales between the neural and plasticity dynamics leads to

$$\tau \dot{J}_i = \mathbf{A}^T \left( \langle n^a \mathbf{x}_i^b \rangle_{n,\mathbf{x}} J_i^c \right) - J_i \sum_j J_j \mathbf{A}^T \left( \langle n^a \mathbf{x}_j^b \rangle_{n,\mathbf{x}} J_j^c \right)$$

(45)

where $\langle n^a \mathbf{x}_i^b \rangle_{n,\mathbf{x}}(t, s)$ is an order $a + b$ joint moment density (correlation function) of the output spike train and the inputs (which might be spike trains or any process admitting a finite joint moment of this order). $\langle \rangle_{n,\mathbf{x}}$ is the expectation over the joint density of the inputs $\mathbf{x}$ and the activity $n$.

In contrast to the original case of eq. 2, these dynamics depend on a joint moment of the inputs and output, rather than on just the input correlation. To calculate this joint moment, we will model the postsynaptic activity as conditionally Poisson. With two additional assumptions, we can recast eq. 45 in a form that depends only on $\boldsymbol{J}$ and statistics of $\mathbf{x}$. First we take the neural transfer function to be a power-law nonlinearity, which matches the effective nonlinearity of mechanistic spiking models in fluctuation-driven regimes [75, 81] and experimental observations [82, 83]. Second, we will assume that the input to the nonlinearity is non-negative, restricting the average over $p(\mathbf{x})$ to one over the samples of $\mathbf{x}$ that can drive spiking.

With the STDP rule of eq. 44, homeostatic regulation of the $p$-norm of the synaptic weights, and a separation of timescales between activity and plasticity, the plasticity dynamics are

$$\tau \dot{J}_i = \mathbf{A}^T\left(\langle n^a \mathbf{x}_i^b\rangle_{n,\mathbf{x}} J_i^c\right) - J_i \sum_j J_j |J_j|^{p-2} \mathbf{A}^T\left(\langle n^a \mathbf{x}_j^b\rangle_{n,\mathbf{x}} J_j^c\right) \tag{46}$$

where for fixed $i$ and $t$ we introduce the inner product over functions:

$$\mathbf{A}^T\left(n^a \mathbf{x}_i^b J_i^c\right)(t) = \int_{-\infty}^{\infty} D\boldsymbol{s}\; A(\boldsymbol{s})\, n(t) \prod_{i=1}^{a-1} n(t+s_i) \prod_{j=a}^{a+b} \mathbf{x}_i(t+s_j) \prod_{k=a+b+1}^{a+b+c} J_i(t+s_k) \tag{47}$$

with integration measure $D\boldsymbol{s} = \prod_{i=1}^{a+b+c} ds_i$. Now we must determine the input-output joint moment $\langle n^a \mathbf{x}_i^b\rangle_{n,\mathbf{x}}$. This will depend on the input distribution, $p(\mathbf{x})$, and the model for the neural activity $n(t)$. We take $n(t)$ to be a Poisson process with stochastic intensity

$$r_{\mathbf{x}}(t) = \phi\left(\boldsymbol{G}^T \mathbf{x}(t) + \lambda(t)\right) \tag{48}$$

where $\boldsymbol{G}(t,s) = \boldsymbol{J}(t) \circ \boldsymbol{W}(s)$ and $\boldsymbol{G}^T \mathbf{x}(t) = \sum_j \int_0^{\infty} ds\; \boldsymbol{G}_j(t-s)\mathbf{x}_j(s)$. That is, $\boldsymbol{J}$ is a vector of synaptic weights and $\boldsymbol{W}$ is a vector of coupling kernels for each synapse. We fix the integral of each elements of $\boldsymbol{W}$ at 1, so $\boldsymbol{J}$ sets the amplitude of synaptic interactions. $\lambda(t)$ models a deterministic drive. We assume that $\boldsymbol{W}$ is fixed and plasticity only affects the weights, $\boldsymbol{J}$. We will also assume that $\boldsymbol{G}^T \mathbf{x}(t) + \lambda(t) \geq 0$.

Our strategy to compute the joint moment $\langle n^a \mathbf{x}_i^b\rangle_{n,\mathbf{x}}$ has two parts. First, we decompose the joint moment into cumulants. Second, we write each of those cumulants as a tensor product of $\boldsymbol{J}$ and a cumulant of $\mathbf{x}$. Only the second step depends on the neuron model.

The joint moment $\langle n^a \mathbf{x}^b\rangle$ can be decomposed into a Bell polynomial in its cumulants:

$$\left\langle n(t) \prod_{l=1}^{b} \mathbf{x}_i(t+s_l) \prod_{m=1}^{a-1} n(t+s_{b+m})\right\rangle_{n,\mathbf{x}} = \sum_{\pi \in \Pi} \prod_{(P,Q)\in\pi} \left\langle\!\!\!\left\langle \prod_{\substack{j\in P \\ k\in Q}} n(t+s_j)\mathbf{x}_i(t+s_k)\right\rangle\!\!\!\right\rangle_{n,\mathbf{x}} \tag{49}$$

where $\Pi$ is the set of all partitions of the time lags $(0, s_1, \ldots, s_{a+b-1})$. ($\Pi$ also corresponds to the set of all partitions of the $a$ factors of $n$ and $b$ factors of $\mathbf{x}_i$ appearing in the joint moment. The first lag, 0, corresponds to $n(t)$.) For one such partition $\pi \in \Pi$, each of its blocks $(P,Q)$ contains indices $j, k$ for the time lags corresponding to factors of $n$ or $\mathbf{x}$. In one block $(P,Q)$ of the partition $\pi$, $P$ is the set of indices $j$ correspond to factors of $n$ while $Q$ is the set of indices $k$ corresponding to factors of $\mathbf{x}$.

We will compute the joint expectation by factorizing $p(n, \mathbf{x}) = p(n|\mathbf{x})p(\mathbf{x})$. This will allow us to write a each joint cumulant of $n, \mathbf{x}$ as a tensor product of $\boldsymbol{J}$ and a cumulant of $\mathbf{x}$. Given $\mathbf{x}$, a cumulant of $n$ is

$$\left\langle\!\!\!\left\langle n(t) \prod_{m=1}^{M} n(t+s_m)\right\rangle\!\!\!\right\rangle_{n|\mathbf{x}} = r_{\mathbf{x}}(t) \prod_{m=1}^{M} \delta(s_m) \tag{50}$$

We will take $\phi(x) = \lfloor x\rfloor_+^d$ so a joint cumulant of $\dot{n}, \mathbf{x}$ is

$$\left\langle\!\!\!\left\langle n(t) \prod_{m=1}^{M} n(t+s_{N+m}) \prod_{n=1}^{N} \mathbf{x}_n(t+s_n)\right\rangle\!\!\!\right\rangle_{n,\mathbf{x}} = \left\langle\!\!\!\left\langle \left\langle\!\!\!\left\langle \dot{n}(t) \prod_m \dot{n}(t+s_{N+m})\right\rangle\!\!\!\right\rangle_{n|\mathbf{x}} \prod_n \mathbf{x}_n(t+s_n)\right\rangle\!\!\!\right\rangle_{\mathbf{x}}$$

$$= \left\langle\!\!\!\left\langle \lfloor \boldsymbol{G}^T\boldsymbol{x}\rfloor_+^d(t) \prod_n \mathbf{x}_n(t+s_n)\right\rangle\!\!\!\right\rangle_{\mathbf{x}} \prod_{m=1}^{M} \delta(s_{N+m}) \tag{51}$$

and if $\boldsymbol{G}^T \mathbf{x} \geq 0$ for all $\mathbf{x}$ then $\lfloor \boldsymbol{G}^T \boldsymbol{x} \rfloor_+^d = \sum_\alpha (\boldsymbol{G}^d)_\alpha^T (\mathbf{x}^d)_\alpha$ so

$$\left\langle\!\!\left\langle n(t) \prod_{m=1}^{a-1} n(t+s_{b+m}) \prod_{l=1}^{b} \mathbf{x}_i(t+s_l) \right\rangle\!\!\right\rangle_{n,\mathbf{x}} = \sum_\alpha (\boldsymbol{G}^d)_\alpha^T \left\langle\!\!\left\langle (\mathbf{x}^d)_\alpha \prod_l \mathbf{x}_i(t+s_l) \right\rangle\!\!\right\rangle_{\mathbf{x}} (t, s_1, \ldots, s_b) \prod_{m=1}^{a-1} \delta(s_{b+m})$$

(52)

These expansions of the input-output joint moment have a similar structure to the expansion of arbitrary learning rules (section 2.3) with one main difference: the exponent of the neural transfer function, $d$, also determines the relevant input moments because of the Poisson cumulants of $n$.

For example, take $a = b = 1$. The relevant joint moment $\langle n^a \mathbf{x}_i^b \rangle_{n,\mathbf{x}}$ is

$$\langle n(t) \mathbf{x}_i(t+s) \rangle_{n,\mathbf{x}} = \sum_\alpha (\boldsymbol{G}^T)_\alpha \langle (\mathbf{x}^d)_\alpha \rangle_\mathbf{x}(t) \langle \mathbf{x}_i \rangle_\mathbf{x}(t+s_1) + \sum_\alpha (\boldsymbol{G}^T)_\alpha \langle\!\langle (\mathbf{x}^d)_\alpha(t) \mathbf{x}_i(t+s_1) \rangle\!\rangle_\mathbf{x}$$

$$= \sum_\alpha (\boldsymbol{G}^T)_\alpha \langle (\mathbf{x}^d)_\alpha(t) \mathbf{x}_i(t+s_1) \rangle_\mathbf{x}$$

(53)

where $\langle\!\langle (\mathbf{x}^d)_\alpha \mathbf{x}_i \rangle\!\rangle = \langle (\mathbf{x}^d)_\alpha \mathbf{x}_i \rangle - \langle (\mathbf{x}^d)_\alpha \rangle \langle \mathbf{x}_i \rangle$ denotes the second cumulant of $(\mathbf{x}^d)_\alpha$ and $\mathbf{x}_i$, not a $d+1$-order cumulant of $\mathbf{x}$, since the factor of $\mathbf{x}^d$ arises from the intensity of $n$. For $a = b = 1$, the decomposition of $\langle n^a \mathbf{x}_i^b \rangle_{n,\mathbf{x}}$ reduces to just the inner product of $\boldsymbol{G}^d$ with a $d+1$-order moment of the inputs, evaluated at one set of time lags. The decomposition of $\langle n^a \mathbf{x}_i^b \rangle_{n,\mathbf{x}}$ does not always reduce to just one term like that. As a second example, take a simple triplet STDP rule ($a = 2, b = 1$). The relevant joint moment $\langle n^a \mathbf{x}_i^b \rangle_{n,\mathbf{x}}$ is

$$\langle n(t) n(t+s_2) \mathbf{x}_i(t+s_1) \rangle = \sum_\alpha (\boldsymbol{G}^d)_\alpha^T \langle\!\langle (\mathbf{x}^d)_\alpha(t) \mathbf{x}_i(t+s_1) \rangle\!\rangle_\mathbf{x} \delta(s_2)$$

$$+ \sum_\alpha (\boldsymbol{G}^d)_\alpha^T \langle (\mathbf{x}^d)_\alpha \rangle_\mathbf{x}(t) \delta(s_2) \langle \mathbf{x}_i \rangle_\mathbf{x}(t+s_1)$$

$$+ \sum_\alpha (\boldsymbol{G}^d)_\alpha^T \langle (\mathbf{x}^d)_\alpha \rangle_\mathbf{x}(t) \sum_\beta (\boldsymbol{G}^d)_\beta^T \langle\!\langle (\mathbf{x}^d(t+s_2))_\beta \mathbf{x}_i(t+s_1) \rangle\!\rangle_\mathbf{x}$$

$$+ \sum_\alpha (\boldsymbol{G}^d)_\alpha^T \langle (\mathbf{x}^d)_\alpha \rangle_\mathbf{x}(t+s_2) \sum_\beta (\boldsymbol{G}^d)_\beta^T \langle\!\langle (\mathbf{x}^d)_\beta(t) \mathbf{x}_i(t+s_1) \rangle\!\rangle_\mathbf{x}$$

$$+ \sum_\alpha (\boldsymbol{G}^d)_\alpha^T \langle (\mathbf{x}^d)_\alpha \rangle_\mathbf{x}(t) \sum_\beta (\boldsymbol{G}^d)_\beta^T \langle (\mathbf{x}^d)_\beta \rangle_\mathbf{x}(t+s_2) \langle \mathbf{x}_i \rangle_\mathbf{x}(t+s_1)$$

(54)

We can recognize two moments of the input here, combining the first and second lines and either the third or fourth with the fifth:

$$\langle n(t) n(t+s_2) \mathbf{x}_i(t+s_1) \rangle_{n,\mathbf{x}} = \sum_\alpha (\boldsymbol{G}^d)_\alpha^T \langle (\mathbf{x}^d)_\alpha(t) \mathbf{x}_i(t+s_1) \rangle \delta(s_2)$$

$$+ \sum_\alpha (\boldsymbol{G}^d)_\alpha^T \langle \mathbf{x}^d \rangle_\alpha(t) \sum_\beta (\boldsymbol{G}^d)_\beta^T \langle (\mathbf{x}^d)_\beta(t+s_2) \mathbf{x}_i(t+s_1) \rangle$$

$$+ \sum_\alpha (\boldsymbol{G}^d)_\alpha^T \langle (\mathbf{x}^d)_\alpha \rangle(t+s_2) \sum_\beta (\boldsymbol{G}^d)_\beta^T \langle\!\langle (\mathbf{x}^d)_\beta(t) \mathbf{x}_i(t+s_1) \rangle\!\rangle$$

(55)

where all expectations on the right-hand side are with respect to the input distribution, $p(\mathbf{x})$.

As discussed above, any joint moment $\langle n^a \mathbf{x}^b \rangle_{n,\mathbf{x}}$ can be decomposed into joint cumulants $\langle\!\langle n^a \mathbf{x}^b \rangle\!\rangle_{n,\mathbf{x}}$. Each of those joint cumulants can be expressed as a tensor product of $\boldsymbol{G}$ with a cumulant of $\mathbf{x}$. To isolate the synaptic weights $\boldsymbol{J}$, let $\mathbf{y} = \boldsymbol{W}^T \mathbf{x}$ so $(\boldsymbol{G}^d)_\alpha^T (\mathbf{x}^d)_\alpha = (\boldsymbol{J}^d)_\alpha \big((\boldsymbol{W}^T \mathbf{x})^d\big)_\alpha = (\boldsymbol{J}^d)_\alpha (\mathbf{y}^d)_\alpha$. Since $\mathbf{y} = \boldsymbol{W}^T \mathbf{x}$, joint cumulants of $\mathbf{x}, \mathbf{y}$ are cumulants of $\mathbf{x}$. So we can write any joint cumulant of $n, \mathbf{x}$ as a tensor product of $\boldsymbol{J}$ with a cumulant of $\mathbf{x}$. Using this and the cumulant decomposition of

$\langle n^a \mathbf{x}_i^b J_i^c \rangle$ in the learning dynamics, eq. 45, yields

$$
\begin{aligned}
\tau \dot{J}_i =\mathbf{A}^T &\left( \left( \sum_{\pi \in \Pi} \prod_{(P,Q) \in \pi} \sum_{\alpha} (\boldsymbol{J}^d)_\alpha^T \left\langle\!\!\left\langle \prod_{\substack{j \in P \\ k \in Q}} (\mathbf{y}^d)_\alpha(t) \mathbf{x}_i(t+s_k) \right\rangle\!\!\right\rangle_{n,\mathbf{x}} \delta(t+s_j) \right) \prod_{l=1}^c J_i(t+s_l) \right) \\
&- J_i \sum_j J_j |J_j|^{p-2} \mathbf{A}^T \left( \left( \sum_{\pi \in \Pi} \prod_{(P,Q) \in \pi} \sum_{\alpha} (\boldsymbol{J}^d)_\alpha^T \left\langle\!\!\left\langle \prod_{\substack{k \in P \\ l \in Q}} (\mathbf{y}^d)_\alpha(t) \mathbf{x}_j(t+s_l) \right\rangle\!\!\right\rangle_{n,\mathbf{x}} \delta(t+s_k) \right) \prod_{m=1}^c J_j(t+s_m) \right)
\end{aligned}
$$
(56)

Equation eq. 56 gives the dynamics of $\boldsymbol{J}$ as a function of $\boldsymbol{J}$ and weighted cumulant tensors of the input $\mathbf{x}$. It has, however, a different form than the corresponding dynamics of the non-spiking neuron (eq. 2). First, the right-hand side is given y a sum of products of cumulant tensors *with* $\boldsymbol{J}^{\otimes d}$, rather than just a sum of products of cumulant tensors.

### A.4 Weight-dependent plasticity

Above, we examined the dynamics of the generalized Hebbian rule with no direct weight-dependence ($c = 0$ in eq. 1). In biological plasticity, this might not be the case. Within dendritic branches, spatially clustered and temporally coactive synapses [40] exhibit cooperative plasticity [36, 41]. Multiplicative weight-dependence also stabilizes Hebbian spike timing–dependent plasticity distributions [6, 7, 84]. As a first step towards incorporating these effects, we consider the dynamics of eq. 2 with $c \neq 0$.

In this case, steady states of the plasticity dynamics (eq. 2) are a new kind of tensor decomposition: $\boldsymbol{J}$ is invariant under $\boldsymbol{\mu}$ up to a scaling and elementwise exponentiation. Are these steady states attractors of eq. 2? Unfortunately, the approach we used to prove theorem 1 does not allow us to answer this question. We next outline the impediment.

Assuming $\boldsymbol{\mu}$ is symmetric and odeco, inserting the orthogonal decomposition (eq. 5) and projecting $\boldsymbol{J}$ onto its factors (as in the proof of theorem 1) yields the dynamics for the eigenvector loadings $\boldsymbol{x}$:

$$
\tau \dot{\boldsymbol{x}} = (\boldsymbol{U}\boldsymbol{x})^{\circ c} \circ \boldsymbol{\Sigma} \mathbf{x}^{\otimes a} - \boldsymbol{x} \circ \left( \boldsymbol{x}^T \boldsymbol{U} \right)^{\circ c+1} \boldsymbol{U} \boldsymbol{\Sigma} \mathbf{x}^{\otimes a+1}
\tag{57}
$$

where $\boldsymbol{x}^{\circ c}$ is the elementwise power of $\boldsymbol{x}$ and $\boldsymbol{U}$ is the matrix with columns composed of the orthogonal components of $\boldsymbol{\mu}$ (eq. 5). (Compare this to eq. 9 in the proof of theorem 1.) If $c \neq 0$, the dynamics of the loadings $\boldsymbol{x}$ are not closed but depend on the structure of the factors in $\boldsymbol{U}$. A general analysis of how $\boldsymbol{U}$ impacts the evolution of $\mathbf{x}$ for $c \neq 0$ is beyond the scope of this study. We will instead consider input distributions that impart simple structure to $\boldsymbol{\mu}$ and analyze the fixed points of eq. 2 for them.

In this section we also generalize the learning dynamics to incorporate a constraint on any $p$-norm of the synaptic weight vector, rather than only its Euclidean norm. This introduces a factor of $|J_j|^{p-2}$ into the second right-hand side term of eq. 2 (appendix A.1).

### A.4.1 Diagonal input correlations

We begin by analyzing inputs with constant-diagonal correlations, $\boldsymbol{\mu}_\alpha = \sigma \delta_\alpha$ with $\sigma > 0$. These could arise if at each time $t$, only one synapse can be activated and the remaining inputs are 0. In that case the only nonzero contribution to $\boldsymbol{\mu}$ would be $\langle \mathbf{x}_i^{b+a} \rangle$. It is possible in this case $\boldsymbol{\mu} = \mathbf{0}$, for example if $\mathbf{x}_i \sim \mathcal{N}(0,1)$ and $a + b$ is odd. Then the leading-order contribution to $\dot{J}_i$ would be supralinear in $dt$. In this case eq. 2 reduces to

$$
\frac{\tau}{\sigma} \dot{J}_i = J_i^{a+c} - J_i \sum_j J_j^{a+c-1} |J_j|^p
\tag{58}
$$

We will analyze fixed points of eq. 58 and their stability. If $\boldsymbol{J}$ is a steady state of eq. 58, its Jacobian matrix is

$$
\frac{\tau}{\sigma} \frac{d\dot{J}_i}{dJ_k} = \delta_{ik} \left( (a+c) J_i^{c+a-1} - \sum_j J_j^{a+c+1} |J_j|^{p-2} \right) - (a+c+p-1) J_i J_k^{a+c} |J_k|^{p-2}
\tag{59}
$$

We will see that sparse connectivity, with one synaptic weight at 1 and the rest at zero, is always a stable equilibrium. In addition, sparse connectivity with one weight at $-1$ is stable if $a + c$ is odd. In addition to these fully sparse steady states, we identify partially sparse and uniform-magnitude equilibria and conditions for their stability. We first state and prove these results. Then, we present simulation results showing that even when other stable equilibria exist, the learning dynamics tend to converge to the fully sparse equilibria.

**Theorem 3.** *Let $\boldsymbol{\mu} \in \mathbb{R}^{K \times K \times \ldots \times K} = \sigma \boldsymbol{\delta}$ be a diagonal tensor of order $a + 1$ with all diagonal elements equal to $\sigma$, $\sigma > 0$. Let $a + c = 1$ with $a \geq 1$. Then the $\ell^p$-sphere in $\mathbb{R}^K$ with unit radius is an attracting slow manifold of eq. 2.*

*Proof.* Setting $\dot{\boldsymbol{J}} = 0$ in eq. 58 yields the steady-state requirement

$$\boldsymbol{J}^* = \boldsymbol{J}^* \Big( \sum_j |J_j^*|^p \Big) \tag{60}$$

Whenever $\|\boldsymbol{J}^*\|_p = 1$, $\Big( \sum_j |J_j^*|^p \Big) = 1$ and vice versa. Any $\boldsymbol{J}^*$ with $\|\boldsymbol{J}^*\|_p = 1$ is thus a steady state of eq. 2. If $\|\boldsymbol{J}^*\|_p \neq 1$, the only steady state is $\boldsymbol{J}^* = \boldsymbol{0}$.

We next consider the linear stability to perturbations around an element $\boldsymbol{J}^*$ of the $\ell^p$-sphere. The Jacobian at $\boldsymbol{J}^*$ is of rank one:

$$\frac{d\dot{J}_i}{dJ_k} = -\frac{\sigma p}{\tau} J_i^* \frac{|J_k^*|^p}{J_k^*} \tag{61}$$

with eigenvalues $-(\sigma p / \tau)\hat{\lambda}$, where $\hat{\lambda}$ is an eigenvalue of $\boldsymbol{A}$, $A_{ik} = J_i^* |J_k^*|^p / J_k^*$. The characteristic equation for $\boldsymbol{A}$ is

$$J_i^* \sum_k \frac{|J_k^*|^p}{J_k^*} \hat{v}_k = \hat{\lambda} \hat{v}_i \tag{62}$$

where $\hat{v}$ is an eigenvector of $\boldsymbol{A}$. Matching indices, the eigenvector $\hat{v}$ with unit $\ell^p$-norm is identical to $\boldsymbol{J}^*$ and it has eigenvalue $\hat{\lambda} = 1$. For $a + c = 1$, any $\boldsymbol{J}^*$ on the $p$-sphere is thus a steady state with one Jacobian eigenvalue $-\sigma p / \tau$, corresponding to the eigenvector $\boldsymbol{J}^*$. The remaining $K - 1$ eigenvalues are zero, so the orthogonal complement of $\boldsymbol{J}^*$ is a slow subspace for the linearized dynamics. Each point on the $\ell^p$ $K$-sphere has such a slow subspace. Together, the $\ell^p$ $K$-sphere is a linearly stable slow manifold. Is it globally attracting? Let

$$L = \sum_{i=1}^K |J_i|^p \tag{63}$$

The total derivative of $L$ with respect to time is

$$\frac{dL}{dt} = p \sum_i J_i |J_i|^{p-2} \dot{J}_i = \frac{p\sigma}{\tau} L (1 - L) \tag{64}$$

which has a stable fixed point at $L = 1$ and an unstable point at $L = 0$. The full synaptic weight dynamics thus must admit a globally attracting subspace on the $\ell^p$ $K$-sphere. Those dynamics are symmetric with respect to rotations of the axes, so that subspace must be the full sphere. $\qquad \square$

**Remark.** *Theorem 3 generalizes the corresponding result for Oja's rule that, when the inputs are zero-mean and uncorrelated ($\mu_{i,j} = \sigma \delta_{i,j}$), the $\ell^2$-sphere is a slow manifold of its dynamics. On it, however, the mean-field dynamics of eq. 58 vanish - so a full accounting of the weight dynamics must examine fluctuations.*

To illustrate these results, we simulated the learning dynamics with individually presented, identically distributed (standard normal) inputs. Since at each time point only one input is presented, the input correlation tensors are diagonal. We first examined the classic Oja rule, taking $(a, b, c) = (1, 1, 0)$. As expected, the synaptic weights exhibited random motion (fig. A.4.1a). Their $p$-norm was fixed and synaptic weights initialized off the unit $p$-sphere quickly converged onto it as predicted by Theorem 3 (fig. A.4.1b).

Next we examined a different parameter set with $a + c = 1$: $(a, c) = (2, -1)$. We kept $b = 1$. In this case, we observed the synaptic weights converge to a sparse solution with one nonzero synapse with

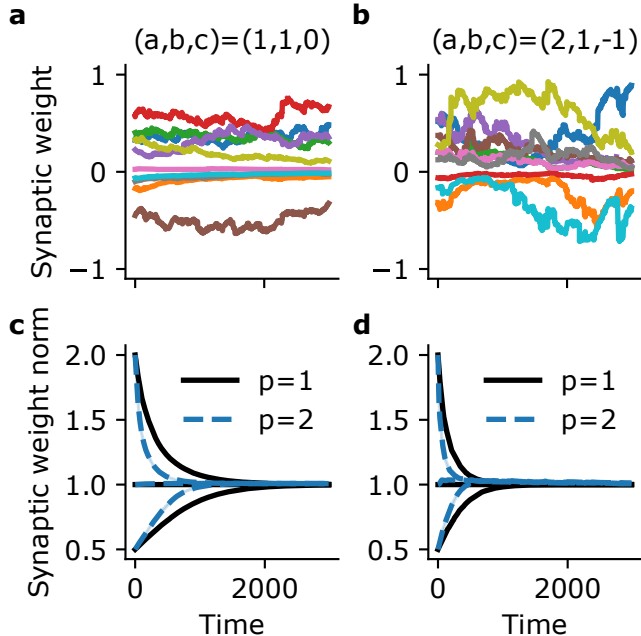

Figure 4: Dynamics of nonlinear Hebbian plasticity rules with weight-dependence and diagonal input correlations: the case $a + c = 1$. For all panels, we used $K = 10$ inputs and a learning rate $\eta = 10^{-2}$. On each time step one uniformly chosen synapse received a normally distributed (mean 1, variance 1) input and the rest had 0 input. **a, b)** Example synaptic weight dynamics with $p = 2$. **c, d)** Norm of the synaptic weight vector. Solid lines: mean over 20 random initial conditions. Shaded areas: standard error. Each curve describes simulations from initial conditions with different norm.

magnitude 1 (fig. A.4.1c). This convergence occurred over a longer timescale than the convergence to the unit sphere for $(a, c) = (1, 0)$. For that previous parameter set, we did not observe synaptic weights converge to these sparse solutions even over this longer timescale (simulation not shown). With $(a, c) = (2, -1)$, the dynamics converged to sparse equilibria for different values of $p$ and for synaptic weights initialized with different variances (fig. A.4.1d). This solution is on the unit $p$-sphere so does not contradict Theorem 3. It is, however, more particular. Next we examine sparse and partially-sparse equilibria, and their stability, for integer-valued $a + c$. We begin by examining even $a + c$, then odd.

**Theorem 4.** *Take $a + c$ even and $\boldsymbol{\mu} \in \mathbb{R}^{K \times K \times \ldots \times K} = \sigma \boldsymbol{\delta}$ be a diagonal tensor of order $a + 1$ with all diagonal elements equal. Let $\{\boldsymbol{J}^* \in \mathbb{R}^K\}$ be the set of $n$-sparse vectors with $n$ nonzero elements $|J_i^*| = n^{-1/p}$. Any such vector where all elements share a sign, $J_i^* = \xi n^{-1/p}$ where $\xi \in \{-1, 1\}$, is a steady state of eq. 58.*

*Proof.* Let $\boldsymbol{J}^*$ be a $n$-sparse vector with nonzero elements $J_i^* = \xi_i n^{-1/p}$, where $\xi_i \in \{-1, 1\}$. Note that with $\xi = \mathbf{1}$, $\boldsymbol{J}^*$ is a steady state solution of eq. 58.

Without loss of generality, permute $\boldsymbol{J}^*$ so that its first $n$ elements are nonzero and last $K - n$ elements are zero. Now, set one element $\xi_i = -1$ and insert this solution for $J$ into the steady-state condition. Since $a + c$ is even, this yields

$$1 - n = \sum_{\substack{j=1 \\ j \neq i}}^{n} \xi_j \tag{65}$$

$\xi_j \in \{-1, 1\}$, so this requires that $\xi = -\mathbf{1}$. If one element of $\xi$ is negative, all must be. The $n$-sparse vector with nonzero elements $J_i^* = -n^{-1/p}$ is also a steady state of eq. 58. $\qquad \square$

**Corollary 4.1.** *If $c = 0$ and $a = 2$, and $\boldsymbol{\mu}$ has finitely many E-eigenvectors, then $\{\boldsymbol{J}^*\}$ contains all the steady states of eq. 2.*

*Proof.* Since $c = 0$, steady state solutions of eq. 2 are also E-eigenvectors of $\boldsymbol{\mu}$. If $\boldsymbol{\mu}$ is a tensor of order 3 with finitely many E-eigenvectors, then it has $2^K - 1$ E-eigenvectors, counted with multiplicity [50, 51].

The set of $n$-sparse vectors with elements $J_i^* = \xi_i n^{-1/p}$, where $\xi \in \{-1, 1\}$, contains steady states of eq. 58. There are

$$\sum_{n=1}^{K} \binom{K}{n} = 2^K - 1$$

such steady states with $\xi = 1$. The corresponding E-eigenvalues are $\lambda = \sum_{j,\alpha} J_j |J_j|^{p-2} \boldsymbol{\mu}_{j,\alpha}(\boldsymbol{J}^2)_\alpha = \sigma \xi n^{-1/p}$. The factor of $\xi$ cancels out in the E-eigenvector/E-eigenvalue equation. So with $c = 0, a = 2$, each of the $n$-sparse steady states with nonzero elements $J_i^* = \xi n^{-1/p}$ is proportional to an E-eigenvector of $\sigma \boldsymbol{\delta}$. Any other vector $\boldsymbol{J}^*$ proportional to an E-eigenvector of $\boldsymbol{\mu}$ would not be a steady state of eq. 58, since the constant of proportionality would obtain a power of 2 in one term of eq. 58 and a power of $2 + p$ in the other term. $\square$

**Theorem 5.** *Take $a + c$ odd and $\boldsymbol{\mu} \in \mathbb{R}^{K \times K \times \ldots \times K} = \sigma \boldsymbol{\delta}$ be a diagonal tensor of order $a + 1$ with all diagonal elements equal. Let $\{\boldsymbol{J}^* \in \mathbb{R}^K\}$ be the set of $n$-sparse vectors with $n$ nonzero elements $J_i^* = \xi_i n^{-1/p}$ where each $\xi_i \in \{-1, 1\}$. Any such $\boldsymbol{J}^*$ is a steady state of eq. 58.*

*Proof.* Let $\boldsymbol{J}^*$ be a $n$-sparse vector with nonzero elements $J_i^* = \xi_i n^{-1/p}$, where $\xi_i \in \{-1, 1\}$. Since $a + c$ is odd, $a + c - 1$ is even and the steady-state condition for $\boldsymbol{J}^*$ is invariant to $\xi$. Each such $\boldsymbol{J}^*$ is a steady state of eq. 58. $\square$

**Corollary 5.1.** *If $c = 0$, $a = 3$, and $\boldsymbol{\mu}$ has finitely many E-eigenvectors, then $\{\boldsymbol{J}^*\}$ contains all the steady states of eq. 2.*

*Proof.* The proof follows the same construction as for Theorem 4.1. Each steady state of eq. 58 corresponds to an E-eigenvector of $\boldsymbol{\mu}$. For $a = 3$, there are (if finitely many) $(3^K - 1)/2$ E-eigenvectors of $\boldsymbol{\mu}$. The set of $n$-sparse vectors with elements $J_i^* = \xi_i n^{-1/p}$, where $\xi_i \in \{-1, 1\}$, contains steady states of eq. 58. There are $\sum_{n=1}^{K} 2^n \binom{K}{n} = 3^K - 1$ such steady states. For each $n$, two of them are equal up to a global sign change which will cancel out with the E-eigenvalue in the E-eigenvalue / E-eigenvector equation. Any other vector $\boldsymbol{J}^*$ proportional to an E-eigenvector of $\boldsymbol{\mu}$ would not be a steady state of eq. 58, since the constant of proportionality would obtain a power of 3 in one term of eq. 58 and a power of $3 + p$ in the other term and $p \geq 1$. So these steady states are all of the weight vectors corresponding to the E-eigenvectors of $\boldsymbol{\mu}$, and they correspond to all of the E-eigenvectors.

$\square$

**Theorem 6.** *Let $\boldsymbol{\mu} \in \mathbb{R}^{K \times K \times \ldots \times K} = \sigma \boldsymbol{\delta}$ be a diagonal tensor of order $a + 1$ with all diagonal elements equal. Let $\{\boldsymbol{J}^*\}$ be the set of $n$-sparse vectors with $n$ nonzero elements and $K - n$ zero elements, with nonzero elements $J_i^* = \xi_i n^{-1/p}$ where $\xi \in \{-1, 1\}$. Let $a + c \neq 1$. Then the vectors in $\{\boldsymbol{J}^*\}$ that are linearly stable steady states of eq. 2 are:*

1. *Fully sparse solutions with one synaptic weight at 1, unless $a + c < 1$*

2. *Fully sparse solutions with one synaptic weight at -1, unless either a) $a + c$ is even and $a + c > 1$ or b) $a + c$ is odd and $a + c < 1$,*

3. *All $n$-sparse vectors with each $\xi_i = 1$, if $a + c = 0$,*

4. *Flat solutions at $\boldsymbol{J} = K^{-1/p}\mathbf{1}$, if $a + c \leq 0$ and even (if $p = 1$ it is marginally stable),*

5. *$n$-sparse solutions with $m \geq 1$ weights at $-n^{-1/p}$ and $n - m$ weights at $n^{-1/p}$, if $a + c < 1$ and odd.*

**Remark.** *If $c = 0$ and $a \in \{2, 3\}$, then $\{\boldsymbol{J}^*\}$ contains all steady states of eq. 58; so the only stable steady states of eq. 58 are those described. Otherwise there might be others.*

*Proof.* We separate the proof into sections describing the different equilibria. We begin with the fully sparse equilibria with one nonzero weight $J_j = \xi$, where $\xi \in \{-1, 1\}$. Fully sparse equilibria. The Jacobian, eq. 59, reduces to

$$\frac{\tau}{\sigma} \frac{d\dot{J}_i}{dJ_k} = -\delta_{ik} \xi^{a+c-1} (a + c - 1 - \delta_{ij}(a + c + p - 1)) \tag{66}$$

where $j$ is fixed. The Jacobian is diagonal and its eigenvalues are $\lambda_1 = -\xi^{a+c-1}(a + c - 1)$, with algebraic multiplicity $K - 1$, and $\lambda_2 = -\xi^{a+c-1}(a + c + p - 1)$. The fully sparse equilibrium with $\xi = 1$ is thus stable unless $a + c < 1$. The fully sparse equilibrium with $\xi = -1$ is unstable if either 1) $a + c$ is odd $a + c < 1$ or 2) $a + c$ is even and $a + c > 1$. The opposite conditions guarantee stability. If $a + c = 1$ the sparse solution is neutrally stable.

Now let the first $1 < n \leq K$ weights be nonzero and $J_j = \xi_j n^{-1/p}$, $j = 1, \ldots, n$. The $n$-sparse solution has Jacobian

$$\frac{\tau}{\sigma} \frac{d\dot{J}_i}{dJ_k} = -\delta_{ik} n^{-(a+c+p-1)/p} \left( \sum_{j=1}^{n} \xi_j^{a+c-1} \right)$$
$$+ \theta(n - i)\theta(n - k) \left( \delta_{ik}(a + c)\xi_i^{a+c-1} n^{-(a+c-1)/p} - (a + c + p - 1)n^{-(a+c+p-1)/p}\xi_i\xi_k \right) \tag{67}$$

We will first consider the case when $a + c$ is even and then when $a + c$ is odd.

Partially sparse and flat equilibria: $a + c$ even. In this case, all $n$ nonzero weights have the same sign, $\xi$, and

$$\frac{\tau}{\sigma} \frac{d\dot{J}_i}{dJ_k} = -\xi\delta_{ik} n^{-(a+c-1)/p} + \theta(n-i)\theta(n-k) \left( \delta_{ik}(a + c)\xi n^{-(a+c-1)/p} - (a + c + p - 1)n^{-(a+c+p-1)/p} \right) \tag{68}$$

where $\theta(x)$ is the Heaviside step function. The Jacobian is the sum of a diagonal matrix and a block-constant matrix. It is similar to a block-diagonal matrix of the form

$$\begin{pmatrix} z e_n e_n^T & 0 \\ 0 & 0 \end{pmatrix} + \begin{pmatrix} x I_n & 0 \\ 0 & y I_{K-n} \end{pmatrix} \tag{69}$$

where $I_q$ is the $q \times q$ identity matrix and $e_n = (1, 0, \ldots, 0)$, and the Jacobian eigenvalues are

$$\frac{\tau}{\sigma}\lambda_1 = x + z = (a + c)n^{-(a+c-1)/p} \left( \xi - n^{(p-1)/p} \right) - (p - 1)n^{-(a+c+p-1)/p},$$
$$\frac{\tau}{\sigma}\lambda_2 = x = \xi n^{-(a+c-1)/p}(a + c), \text{ with algebraic multiplicity n} - 1 \tag{70}$$
$$\frac{\tau}{\sigma}\lambda_3 = y = -\xi n^{-(a+c-1)/p}, \text{ with algebraic multiplicity K} - n$$

If $1 < n < K$, the latter two eigenvalues guarantee instability whether $a + c > 0$ or $a + c < 0$, since they share $\xi = \pm 1$. Let $n = K$, so $\lambda_3$ doesn't exist. In this case,

$$\lambda_1 = K^{-(a+c+p)/p} \left( (a + c) \left( \xi K^{(p+1)/p} - K^2 \right) + K^{1/p}(1 - p) \right) \tag{71}$$

and $\lambda_1$ is negative if

$$(a + c) \left( K^{2-1/p} - \xi K \right) < 1 - p \tag{72}$$

We can determine the behavior of $\lambda_1$ by recalling that $p \geq 1$ so $K^{2-1/p} \geq K$ with equality at $p = 1$. If $p = \xi = 1$, then $\lambda_1 = 0$ and the flat equilibrium has an associated slow direction. The equilibrium, $J = K^{-1/p}\mathbf{1}$, is then marginally stable if $\lambda_2 \leq 0$, which occurs when $a + c < 0$.

If $p > 1$ and $a + c > 0$ then $\lambda_1 > 0$ for any $K$ whether $\xi = 1$ or $\xi = -1$. If $p > 1$ and $a + c < 0$, then $\lambda_1 < 0$ for either sign of $\xi$. In that case, $\lambda_2 < 0$ only if $\xi = 1$. So for $p > 1$ and even $a + c$, the uniform steady states with $\xi = 1$ is stable if $a + c < 0$ and unstable if $a + c > 0$.

If $a + c = 0$, $\lambda_2 = 0$ and there are $n - 1$ slow directions associated with each $n$-sparse equilibrium (since, in the basis of eq. 69, these eigenvalues are associated with the unit basis eigenvectors).

Inspection of $\lambda_1, \lambda_3$ reveals that $n$-sparse equilibria are linearly stable with $\xi = 1$ and unstable with $\xi = -1$.

Partially sparse and flat equilibria: $a + c$ odd. Let $n \geq 2$. Without loss of generality, let the first $0 \leq m \leq n$ nonzero weights be negative, the next $n - m$ weights be positive, and the remaining $K - n$ weights be 0. The Jacobian is

$$\frac{\tau}{\sigma}\frac{d\dot{J}_i}{dJ_k} = -\delta_{ik}n^{-(a+c-1)/p} + \theta(n-i)\theta(n-k)\left(\delta_{ik}(a+c)n^{-(a+c-1)/p} - (a+c+p-1)n^{-(a+c+p-1)/p}\xi_i\xi_k\right)$$
(73)

which is a sum of block-diagonal and block-constant matrices,

$$\begin{pmatrix} xI_n & 0 \\ 0 & yI_{K-n} \end{pmatrix} + \begin{pmatrix} C & 0 \\ 0 & 0 \end{pmatrix}$$
(74)

where $C$ is a $n \times n$ block matrix, with entries $C_{ik} \propto \xi_i\xi_k$. We can calculate the eigenvalues of $C$ by noticing that it is the sum of constant and diagonal matrices. The final Jacobian eigenvalues are

$$
\begin{aligned}
\frac{\tau}{\sigma}\lambda_1 &= n^{-(a+c-1)/p}\left(a+c-1\right), \\
\frac{\tau}{\sigma}\lambda_{2+} &= n^{-(a+c-1)/p}\left(a+c-1\right)\left(1+(p/n)\left(m-(n-m)+2+\sqrt{(n-2)^2+4m(n-m)}\right)/2\right), \\
\frac{\tau}{\sigma}\lambda_{2-} &= n^{-(a+c-1)/p}\left(a+c-1\right)\left(1+(p/n)\left(m-(n-m)+2-\sqrt{(n-2)^2+4m(n-m)}\right)/2\right), \\
\frac{\tau}{\sigma}\lambda_3 &= n^{-(a+c-1)/p}\left(a+c-1\right)\left(1+2p/n\right), \\
\frac{\tau}{\sigma}\lambda_4 &= -n^{-(a+c-1)/p}, \text{ exists if } n < K
\end{aligned}
$$
(75)

If $a + c = 1$, these are all zero except $\lambda_4$ which is negative. Take $a + c \neq 1$ and odd. $a + c$ might be positive or negative. If $a + c > 1$, $\lambda_1$ and $\lambda_3$ guarantee instability. If $a + c < 1$ then $\lambda_1, \lambda_3, \lambda_4$ are all negative and the only possible instability is in $\lambda_2\pm$. The discriminant appearing inside the square root in $\lambda_2\pm$, $D$, is strictly increasing with respect to $n$. Take $\lambda_{2+}$. If $a + c < 1$, then for fixed $n$ it is maximized at $m = 0$:

$$\lambda_{2+} \leq \frac{\sigma}{\tau}n^{-(a+c-1)/p}\left(a+c-1\right)\left(1+\frac{p}{2n}\right) < 0$$
(76)

so $\lambda_{2+} < 0$ and $\lambda_{2-}$ determines the stability. If $a + c < 1$ then for fixed $n$, $\lambda_{2-}$ is also maximimized at $m = 0$:

$$\lambda_{2-} \leq \frac{\sigma}{\tau}n^{-(a+c-1)/p}\left(a+c-1\right)\left(1+2\frac{p}{n}-p\right)$$
(77)

If $a + c < 1$ and $p = 1$, that upper bound is always negative. If instead $p > 1$ and $n < 2p/(p-1)$, then the upper bound for $\lambda_{2-}(m)$ is positive: as long as $m$ is sufficiently small, $\lambda_{2-}$ can be positive. $\lambda_{2-}$ is negative if

$$m > \frac{n(1-p)+\sqrt{n^2(p^2-1)+2p^2(1-n)}}{2np}$$
(78)

and $\lambda_{2-}$ is positive if the inequality is reversed. That bound is less than or equal to

$$0 < \frac{1-p+\sqrt{p^2-1}}{2p} < 1$$
(79)

and approaches it from below as $n \to \infty$. So for $a + c < 1$ and odd (i.e., negative) at least one negative synaptic weight is required to stabilize a $n$-sparse steady state. $\qquad\square$

We have constructed a number of steady states for the nonlinear Hebbian dynamics with weight dependence and examined conditions for their stability. If $c \neq 0$ and $a + c \neq 1$, there are always $K$ stable sparse equilibria. In several cases, there are also other stable equilibria also (theorem 6). eq. 2 is a limiting deterministic description (large $\tau$) of an underlying stochastic dynamics, eq. 3. Here we asked whether the fixed points we described above accurately describe the stochastic system. To examine the learning dynamics with diagonal input correlations, we presented i.i.d inputs to one synapse at a time. Since at each time point only one input is presented, the input correlation tensors

are diagonal. We examined parameter sets in each of the cases of theorem 6. For odd $a + c > 0$, the only stable $n$-sparse equilibria are fully sparse with one weight at 1 or -1 (fig. A.4.1a). These were also the only equilibrium we observed over 50 randomly chosen initial conditions (fig. A.4.1b). For even $a + c > 0$, the only stable equilibrium described in theorem 6 is fully sparse with one weight at 1. For such parameters, that was the only equilibrium we observed (fig. A.4.1c, d).

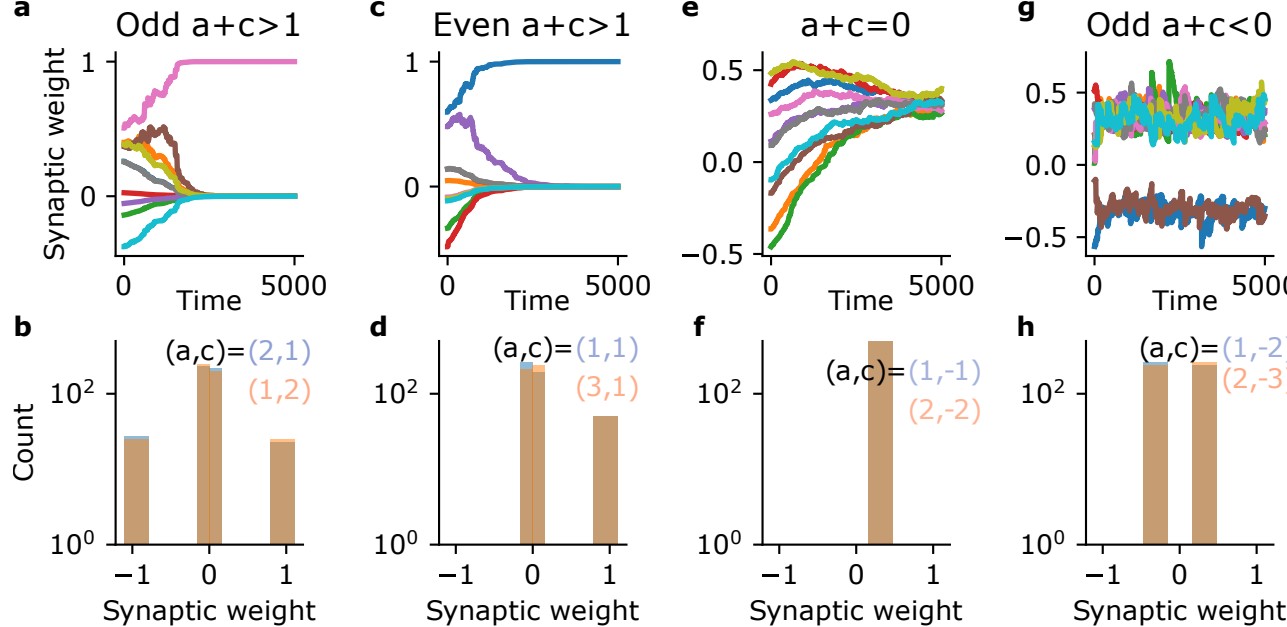

Figure 5: Dynamics of nonlinear Hebbian plasticity rules with weight-dependence and diagonal input correlations. For all panels, we used $K = 10$ inputs. The learning rate was $\eta = 10^{-2}$ for all panels except **e-h**, which had $\eta = 10^{-3}$. **a)** Convergence to the sparse solution with one $J_i = 1$ for $a + c > 0$ and odd. **b)** Histogram of final synaptic weight values after $T = 10^3$ time steps, across 50 random initial conditions. Synaptic weights were averaged over the final 500 time points to smooth out fluctuations for visualization. **c)** Convergence to the sparse solution with one $J_i = 1$ for $a + c > 0$ and even. **d)** Histogram of final synaptic weight values (as in panel **a**). **e)** Convergence to the flat solution at $\boldsymbol{J} = K^{-1/p}\boldsymbol{1}$ for $a + c = 0$. **f)** Histogram of final synaptic weight values (as in panel **a**). **g)** Convergence to a bimodal distribution with 2 synaptic weights at $-K^{-1/p}$ and the remaining 8 at $K^{-1/p}$. **h)** Histogram of final synaptic weight values (as in panel **a**).

For $a + c = 0$, theorem 6 describes a combinatorial explosion of equilibria: each of the $n$-sparse steady states is stable. There are $\sum_{n=1}^{K} \binom{K}{n} = 2^K - 1$ such points, each with $n - 1$ neutrally stable directions. In simulations, we only observed convergence to the flat solution with $n = K$ and all weights at $K^{-1/p}$ (fig. A.4.1e, f). The stochastic dynamics we simulated contain terms proportional to $J_i^c$; this is the origin of the powers of $c$ in eq. 58. Since $c < 0$ these factors explode for $J_i \to 0$. So the only partially sparse solution consistent with the stochastic dynamics is the one with $n = K$ nonzero weights.

Finally, for $a + c < 0$, theorem 6 describes an even greater combinatorial explosion of equilibria. Each $n$-sparse steady state with $1 < m < n$ negative weights and $n - m$ positive weights is linearly stable. There are $\sum_{n=1}^{K} \binom{K}{n} \sum_{m=1}^{n} \binom{n}{m} = 3^K - 2^K$ such equilibria. As before, however, if any $J_i \to 0$ the stochastic dynamics would explode because of the factors $J_i^c$. ($a + c < 0$ requires $c < 0$ since $a > 0$ by assumption.) So again, we see that the only possible steady states for the stochastic dynamics have $K$ nonzero weights (fig. A.4.1g, h). In this case there are $\binom{K}{m}$ equilibria with $m$ negative synaptic weights and $\sum_{m=1}^{K} \binom{K}{m} = 2^K - 1$ such equilibria in total. With odd $a + c < 0$, any of these are stable and we observed convergence to various of them (fig. A.4.1g, h). For even $a + c < 0$, only the flat solution with all weights at $K^{-1/p}$ are linearly stable. In simulations, we did

not observe convergence to this solution. Instead we observed large fluctuations characterized by prolonged excursions of individual synaptic weights (fig. A.4.1a, b). When $a + c \neq 1$, the dynamics of the synaptic weight norm are not closed. With $a + c < 0$ and even, the unit-norm $\ell^p$-sphere appeared unstable in simulations (fig. A.4.1c, d).

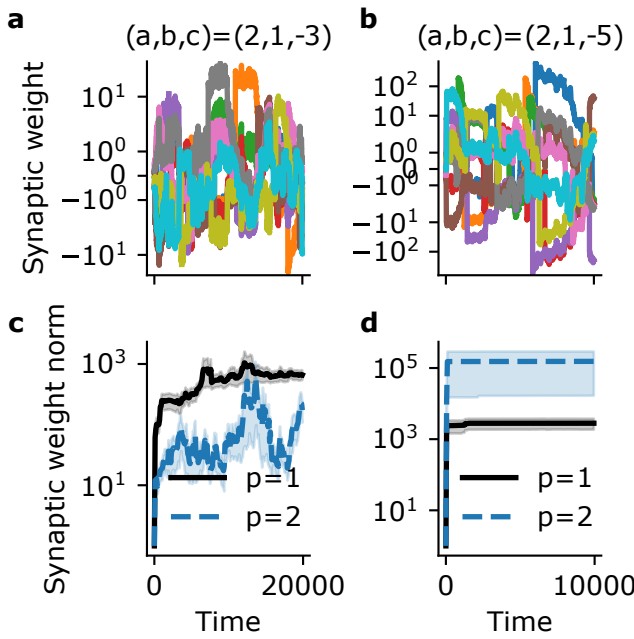

Figure 6: Large fluctuations in synaptic weights for $a + c < 0$ and even. **a)** Example dynamics for two different parameter sets. **b)** Evolution of the synaptic weight norm over 20 realizations. **c)** Impact of decreasing the learning rate.

### A.4.2 Rank one input correlations

Let $\boldsymbol{\mu} = \boldsymbol{r}^{a+1}$, the $(a+1)$-fold outer product of the vector $\boldsymbol{r}$. This corresponds to the case of constant inputs. The dynamics reduce to

$$\tau \dot{J}_i = r_i J_i^c (\boldsymbol{r}^T \boldsymbol{J})^a - J_i (\boldsymbol{r}^T \boldsymbol{J})^a \sum_j r_j J_j^{c-1} |J_j|^p \tag{80}$$

and at steady states,

$$r_i J_i^c (\boldsymbol{r}^T \boldsymbol{J})^a = J_i (\boldsymbol{r}^T \boldsymbol{J})^a \sum_j r_j J_j^{c-1} |J_j|^p \tag{81}$$

Weights orthogonal to the input direction, $\boldsymbol{r}^T \boldsymbol{J} = 0$, are a steady state. Otherwise, we see that $J_i = 0$ is always a steady state for $c > 0$. If $r_i = 0$, then either $J_i = 0$ or $\sum_j r_j J_j^{c-1} |J_j|^p = 0$. If $\boldsymbol{J}$ is a steady state, the Jacobian is

$$\tau \frac{d\dot{J}_i}{dJ_k} = \delta_{ik}(rJ)^a \left( cJ_i^{c-1} r_i - J_j^{c-1}|J_j|^p r_j \right) + a(rJ)^{a-1} \left( J_i^c r_i - J_i J_j^{c-1}|J_j|^p r_j \right) r_k - (c+p-1)(rJ)^a J_i J_k^{c-1}|J_k|^p r_k \tag{82}$$

where $(\boldsymbol{r}^T \boldsymbol{J})^0 = 1$, including at $\boldsymbol{r}^T \boldsymbol{J} = 0$. At an orthogonal steady state, $\boldsymbol{r}^T \boldsymbol{J} = 0$, the Jacobian simplifies to exactly 0 so that direction defines a slow subspace of the linearized dynamics.

By definition, $\boldsymbol{r}$ is an E-eigenvector of $\boldsymbol{\mu}$ with eigenvalue $\|\boldsymbol{r}\|_2^{2a}$ and $\boldsymbol{\mu}$ has a rank one CP decomposition in $\boldsymbol{r}$. So if $(b, c) = (1, 0)$, $\boldsymbol{J} = \boldsymbol{r}$ is an attracting steady state of eq. 80 (theorem 1). Here we focus on the dynamics with weight-dependence. We study the simple case of $c = 1$ and a piecewise constant $\boldsymbol{r}$ with $n$ elements equal to $r$, and the remaining zero. We see that in this case, the unit-norm $n$-sphere is an equilibrium set for the dynamics and determine when it is stable.

**Theorem 7.** *Let $\boldsymbol{\mu} \in \mathbb{R}^{K \times K \times \dots \times K} = \boldsymbol{r}^{a+1}$ be a rank one tensor of order $a+1$, the $(a+1)$-fold outer product of $\boldsymbol{r}$, where $\boldsymbol{r} \in \mathbb{R}^K$. Let $\boldsymbol{r}$ be $n$-sparse and piecewise constant, with $n$ nonzero elements equal to $r$ and the remaining $K - n$ elements zero. Let*

$$M(\boldsymbol{J}) = \sum_{i=1}^{K} J_i \tag{83}$$

*and name $\mathcal{S}$ the unit $\ell^p$ $n$-sphere in $\mathbb{R}^K$, with nonzero elements on the dimensions corresponding to the nonzero elements of $\boldsymbol{r}$. If $c = 1$ then*

1. *The $K - n$ elements of $\boldsymbol{J}$ corresponding to the zero elements of $\boldsymbol{r}$ have a fixed point at zero. It is stable if $M(\boldsymbol{J}) > 0$ and unstable if $M(\boldsymbol{J}) < 0$.*

2. *$\mathcal{S}$ is a slow manifold for the dynamics of the remaining $n$ synaptic weights. It is stable if $a$ is odd or $r > 0$ and unstable if both $a$ is even and $r < 0$.*

*Proof.* Let $\boldsymbol{r}$ be $n$-sparse and piecewise constant, with its first $n$ elements equal to $r$ and the remaining $K - n$ elements zero. Assume that $\boldsymbol{J} \neq 0$. We proceed in order of the claims. First consider the $K - n$ inputs where $r_i = 0$. For $c = 1$, their dynamics are

$$\tau \dot{J}_i = -r^{a+1} M^a(\boldsymbol{J}) L(\boldsymbol{J}) J_i \tag{84}$$

where

$$L(\boldsymbol{J}) = \sum_{i=1}^{K} |J_i|^p \tag{85}$$

$L \geq 0$ by definition with equality only at $\boldsymbol{J} = 0$. So if $r^{a+1} M^a > 0$, these weights will converge to a steady state at zero. If $r^{a+1} M^a < 0$, these weights will diverge exponentially. If $M$ is fixed at $0$ these weights are stable.

Second consider the dynamics of the $n$ weights with nonzero $r_i$, which reduce to

$$\tau \dot{J}_i = r^{a+1} M^a(\boldsymbol{J}) \left(1 - L(\boldsymbol{J})\right) J_i \tag{86}$$

and the steady state condition for $J_i$ is that either $J_i = 0$, $M = 0$ or $L = 1$. So we have steady states for the first $n$ elements of $\boldsymbol{J}$ on either the $\ell^p$ $n$-sphere or on the hyperplane orthogonal to $\boldsymbol{1}$ (and the trivial steady state $J_i = 0$). Next we examine stability for those $n$ weights at one such point $\boldsymbol{J}^*$. From eq. 82, the Jacobian matrix at $\boldsymbol{J}^*$ has rank one

$$\tau \frac{d\dot{J}_i}{dJ_k} = -p r^{a+1} M^a(\boldsymbol{J}^*) J_i^* |J_k^*|^p \tag{87}$$

It has one eigenvalue $-(p/\tau) r^{a+1} M^a(\boldsymbol{J}^*) \sum_j J_j |J_j^*|^p$, with associated eigenvector $\boldsymbol{J}^*$. The remaining $n - 1$ eigenvalues are zero, and the orthogonal complement of $\boldsymbol{J}^*$ is their slow eigenspace. Each point $\boldsymbol{J}^*$ on the $\ell^p$ $n$-sphere has such a slow eigenspace so the full sphere is a slow manifold. To determine the stability of the unit-norm $n$-sphere we will examine the dynamics of the synaptic weight norm. The dynamics of $L$ and $M$ form a closed system:

$$\begin{aligned} \tau \dot{L} &= p r^{a+1} M^a L (1 - L) \\ \tau \dot{M} &= r^{a+1} M^{a+1} (1 - L) \end{aligned} \tag{88}$$

There are two line equilibria on $M = 0$ and $L = 1$ and the Jacobian determinant is $p r^{a+3} M^{a+1} (1 - L)^2$, which is zero on either of those line equilibria so a linear stability analysis is uninformative. Recall that $L \geq 0$ by definition. There are three relevant cases for the dynamics. When $a$ is odd, all factors of $r$ are positive and so is $M^{a+1}$. When $a$ is even, the sign of $r$ impacts the sign of $\dot{M}$. We next examine the three cases: 1) $a$ odd, 2) $a$ even and $r > 0$ and 3) $a$ even and $r < 0$.

First take $a$ odd (fig. A.4.2a). Then $L = 1$ is attracting when $M > 0$ but repelling when $M < 0$. $M$ is always increasing for $L < 1$ and decreasing for $L > 1$. With $a$ even and $r > 0$, $L = 1$ is always attracting (fig. A.4.2b). $M = 0$ is attracting for $L > 1$ and vice versa. If $a$ is even and $r < 0$, $L = 1$ is always repelling. In this case, if $L(0) > 1$ the synaptic weights will explode while if $L(0) < 1$ the synaptic weights will evolve towards the stable equilibrium $L = 0, M = 0$ (fig. A.4.2c). This corresponds to $\boldsymbol{J} = 0$. In sum, the unit-norm solution $L = 1$ can be attracting or repelling. It is attracting if $a$ is odd, or $a$ even with $r > 0$. It is repelling if $a$ is even and $r < 0$. $\square$

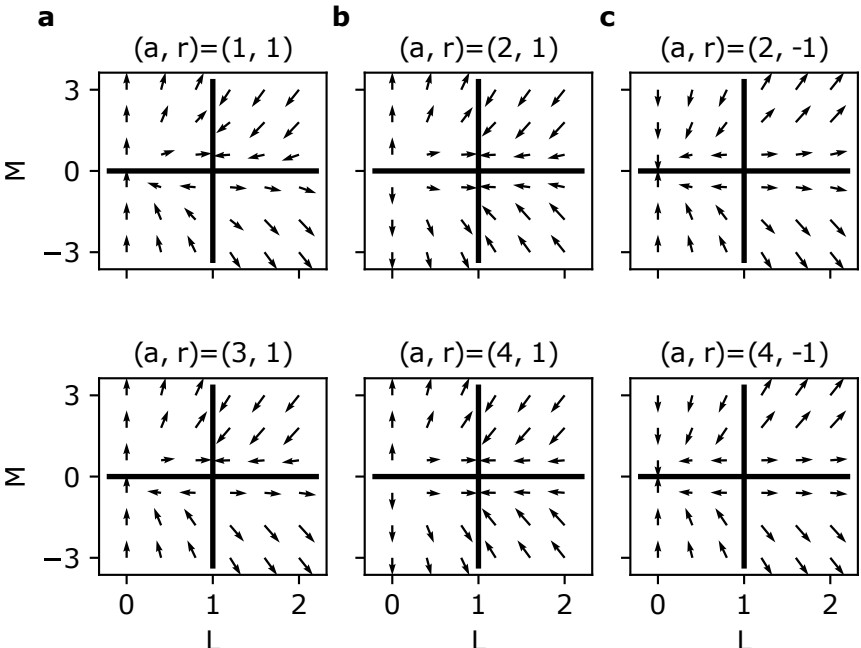

Figure 7: Dynamics of the synaptic weight norm (phase portraits). The vectors $\dot{L}, \dot{M}$ are plotted with unit norm. For each case, we show two corresponding parameter sets. **a)** Case 1: $a$ odd. **b)** Case 2: $a$ even and $r > 0$. **c)** Case 3: $a$ even and $r < 0$.