# OpenReview forum: "Tensor decompositions of higher-order correlations by nonlinear Hebbian plasticity"
_NeurIPS.cc/2021/Conference — NeurIPS 2021 Poster_

### Official Review · Reviewer_gbB7 · 2021-07-13

**Rating:** 6
**Confidence:** 5

**Summary:**

This work proposes to determine the solutions of "generalized" Hebbian rules, i.e., applied to tensor matrices, and analyze the stability of these solutions. The model starts by postulating a biologically sensible form of nonlinear Hebbian rule as polynomials of local input activity, postsynaptic activity, and current synaptic weight. From this generalized form of Hebbian rule, the authors can determine the nature of the steady-states of the dynamic and relate it to the spectrum and eigenspace of the higher-order covariance matrix. Furthermore, the size of the basins of attraction is determined. The model is then used on toy experiments proving the convergence of the learning rules for specifically chosen parameters of the learning rule.

**Limitations And Societal Impact:**

The limitations of the paper are hard to find as the paper does not clearly contextualize in which fields it operates, as mentioned in the main review.

**Main Review:**

First is addressed the organization and clarity of the paper.

The paper is well-written, and results and theorems are nicely explained using informal explanations and leaving the full proof to the appendix. Unfortunately, the paper starts with a "Result" section without introducing the necessary background and existing rules and contextualizing the problem. Related work can certainly appear at the end, but I believe various things should have been introduced early on.

For example:
1) What are the limitations of existing Hebbian rules, and why do we need higher-order power? And what is the difference with nonlinear inputs/outputs?

2) The point above would have naturally lead to the introduction of higher-order covariance matrices, which are naturally formulated as tensors. As a result, one can regret the mention of prevalent tasks that is tensor PCA or independent component analysis (very briefly mentioned by the authors). For example, the work of Cichocki, most recently in [i], presents examples where tensor decomposition is essential. The author of [i] is one of many interested in biological implementation for third and fourth-order tensors and has been omitted to be cited. For example, the work of [ii] proposes similar learning rules in the case of ICA and discusses the possible biological implementation using higher-order neurons.

3) Tensor decomposition is also a very active field of research in the machine learning community, and one can regret that "gradient-based" methods (which Oja is one of) have not been discussed.

The experiments are following each other as they happened chronologically, following the authors' discoveries as they happened rather than proposing an overarching methodology. It can be seen when using "We next investigated" "We next asked" followed by another "We next investigated" in section 2.3. Having a separate methodology section could help the reader understand what the paper aims to achieve. But it still important to point out that paragraphs taken separately are nicely written and clear.

Now is addressed the originality of the paper.

The main originality of the paper is the analysis of the solution of the ad-hoc learning rule. It cannot be said that such learning has never been proposed nor analyzed in other contexts cf reference given above. Hebbian/Oja's rules have long been known to be gradient-based methods of "Extended covariance" matrices, so this result is not very surprising, although important to be theoretically proved. Unfortunately, it is also true that how these rules can be biologically implemented is heavily discussed and shown to require recurrent connections or to operate on various timescales when considering more than one output neuron. These facts are largely overlooked, which greatly weakens the paper as its relevance can be questioned from the start if the main assumptions of the paper are not properly cleared out.
The authors claim to address phenomenological rules, but no results seem to validate or invalidate known plasticity rules. One could have preferred fewer but clearer examples that would relate this work to neuroscientific facts, rather than speculations of what is and what is not biologically plausible. In the case of the spiking neuron (Sec 3.1), this work should have related to the paper from [iv] which would have strengthened the paper as it offers a nice theoretical framework for it.

Now is addressed the significance of the paper.

The paper appears to propose an interesting analysis of the solution of an ad-hoc learning rule. Unfortunately, this work does not make predictions on how (implementation-wise) and where (structurally) they could occur in the brain. There have also been various higher-order neuron models that have been used c.f. [ii] in concrete tasks, when the experiments are not really relatable and do not add much to the paper. One could have expected examples where tensor decomposition has been used either in neuroscience or machine learning.



[i] Cichocki, Andrzej. "Tensor decompositions: a new concept in brain data analysis?." arXiv preprint arXiv:1305.0395 (2013).

[ii] Ziegaus, Ch, and Elmar Wolfgang Lang. "A neural implementation of the JADE algorithm (nJADE) using higher-order neurons." Neurocomputing 56 (2004): 79-100.

[iii] Ge, Rong, et al. "Escaping from saddle points—online stochastic gradient for tensor decomposition." Conference on learning theory. PMLR, 2015.

[iv] Gjorgjieva, Julijana, et al. "A triplet spike-timing–dependent plasticity model generalizes the Bienenstock–Cooper–Munro rule to higher-order spatiotemporal correlations." Proceedings of the National Academy of Sciences 108.48 (2011): 19383-19388.


**Time Spent Reviewing:**

4

---

> ### Author Response · Authors · 2021-08-06
> **review response**
>
> Thank you for your review. We’re glad you found the paper well-written, and appreciate your suggestions for improving it and references to related work, which we now include.
>
> In response to your and the other reviewers’ comments, we have prepared a revised introduction and related work section, which we copy below. The introduction now includes a paragraph discussing applications of tensor decompositions (including the references you suggested, as well as Kolda & Baker's review, the book "Multi-way Array (Tensor) Factorizations and Decompositions" by Cichocki, Zdunek, Phan & Amari, the Cichocki arXiv paper you referenced, and Williams et al. 2020 for a recent example in neuroscience.
>
> We hope these new/revised sections help motivate and contextualize our work. We are open to moving the related work section into the introduction as below (or otherwise revising them) if the reviewers think that would improve the manuscript. If you have any other specific instructions for improving the introduction, we would welcome them.
>
> Regarding biological motivation: The biological findings motivating our family of plasticity rules were discussed in the second paragraph of the introduction. Those experiments were carried out in the sensory cortices of mice and rats. To clarify this, we have edited the first sentence of that paragraph to specify that the basic Hebbian postulate does not account for fundamental aspects of plasticity in cortical pyramidal neurons.
>
> Regarding originality: The most directly related work that we are aware of is that of Taylor & Coombes, who studied an adaptation of Oja’s rule to higher-order neurons: neurons with synaptic weights that provide specific coupling to coincident inputs, e.g., $n = \sum_{j,k,l} W_{jkl} \mathrm{x}_j \mathrm{x}_k \mathrm{x}_l$. This is a completely different model to the one we study, where the synaptic weights only couple to one input: $\bf{J}$ is a vector in our model. This is discussed in the new related work section.
>
> Thank you for the reference to the nJADE paper showing that these higher-order neurons can be used to learn ICA; we weren’t aware of it, and now cite it along with Taylor & Coombes in the related work section.
>
> Revised introduction and related work section:
>
> **Introduction**
>
> The strength of synaptic connectivity depends on pre- and postsynaptic activity. In Hebbian learning rules, potentiation of the net synaptic weight between two neurons is driven by the correlation between pre- and postsynaptic activity [1]. That postulate is a cornerstone of the theory of synaptic plasticity and learning [2, 3]. In its basic form, the Hebbian model leads to runaway potentiation or depression of synapses, since the pre-post correlation increases with increasing synaptic weight [4]. That runaway potentiation can be stabilized by supplemental homeostatic plasticity dynamics [5], by weight dependence in the learning rule [6, 7], or by synaptic scaling regulating a neuron’s total synaptic weight [8, 9]. In 1982, Erkki Oja observed that a linear neuron with Hebbian plasticity and synaptic scaling learns the first principal component of its inputs [10]. This led to a fountain of research on unsupervised learning in neural networks [11, 12].
> The basic Hebbian postulate does not take into account fundamental nonlinear aspects of biological synaptic plasticity in cortical pyramidal neurons. First, synaptic plasticity depends on beyond- pairwise activity correlations [13–18]. Second, spatially clustered and temporally coactive synapses exhibit correlated and cooperative plasticity [19–26]. There is a rich literature on computationally motivated forms of nonlinear Hebbian learning (see 1.1). Here, we will prove that these biologically motivated nonlinearities allow a neuron to learn higher-order features of its inputs.
>
> Principal component analysis (PCA) describes second-order features of a random variable. Both naturalistic stimuli and neural activity can, however, exhibit higher-order correlations—features that are not described by PCA [27, 28]. Canonical models of retinal and thalamic processing whiten inputs, removing pairwise features [29–35]. Beyond-pairwise features, encoded in tensors, can provide a powerful substrate for learning from data [36–39].
>
> **Related work and applications**
>
> There is a rich literature on generalized or nonlinear forms of Hebbian learning. We briefly discuss the most closely related results, to our knowledge. Learning rules with suitable postsynaptic nonlinearities can allow a neuron to perform independent components analysis (ICA) [40, 41]. These learning rules typically minimize or maximize the kurtosis of the neural response. In contrast, we will show that a simple nonlinear Hebbian model directly finds tensor eigenvectors of a higher-order correlation of the input. Those higher-order input correlations can determine which independent features are learned by gradient-based ICA algorithms [42].
> Taylor & Coombes showed that a generalization of the Oja rule to higher-order neurons can also learn higher-order correlations [43], which can allow learning independent components [44]. That construction relies on a model with synaptic weights that specifically couple the postsynaptic neuron to coincident inputs; the synaptic weights are themselves a higher-order tensor. We focused on
> “first-order" synaptic weights with a subsequent nonlinearity in the plasticity.
> The family of Bienenstock, Cooper & Munro (BCM) learning rules supplement the classic Hebbian model with a stabilizing sliding threshold for potentiation rather than synaptic scaling [45]. BCM rules balance terms driven by third and fourth-order joint moments of the pre- and postsynaptic activity [46]. A triplet STDP model with rate-dependent depression and uncorrelated Poisson spiking has BCM dynamics [47] and can develop selective (sparse) connectivity in response to rate- or correlation-based input patterns [48]. If the input x is drawn from a mixture model then under a BCM rule, the synaptic weights are guaranteed to converge to the class means of the mixture [49].
> From a computational perspective, nonlinear Hebbian learning has attractive properties for tensor decompositions compared to tensor power iteration [50], which requires the full tensor μ to operate on, with O(Ka+1) space complexity (see eqs. 1, 3 for variable definitions; K is the dimensionality of the inputs). Streaming computations that operate on one input sample at a time, such as those of eq. 2, have O(K) space complexity. The discrete-time dynamics of eq. 1 correspond to streaming tensor power iteration for computing tensor singular vectors [51–53]). With those parameters, eq. 2 can thus be viewed as the continuous-time limiting dynamics of tensor power iteration. Convergence proofs for the discrete-time dynamics require a learning rate that decreases over time. Biological mechanisms for such a scheme are unclear.
>
> **Our contribution**
>
> We study the dynamics of a simple family of generalized Hebbian learning rules motivated by these observations discussed above, combined with synaptic scaling (eq. 1). We show that its equilibrium synaptic weights correspond to low-rank tensor decompositions for higher-order input correlations. The order of input correlation (two-point, triplet, etc.) depends on the pre- and postsynaptic nonlinearities of the learning rule. When the only nonlinearity in the plasticity rule is postsynaptic, the steady states are tensor eigenvectors [50, 54, 55] of higher-order input correlations. We prove that the higher-order input correlation tensor’s eigenvectors are attractors of the generalized Hebbian plasticity dynamics and characterize their basins of attraction.
>
> Then, we study further generalizations of these learning rules. We show that any plasticity model (with a finite Taylor expansion in the synaptic input, neural output, and synaptic weight) has steady states that generalize those tensor decompositions to multiple input correlations, including generalized tensor eigenvectors. We show that these generalized tensor eigenvector are stable equilibria of the learning dynamics. Due to the complexity of the arbitrary learning rules, we are unable to fully determine their basins of attraction. We do find that they are contained in an attracting set for the dynamics, and characterize its basin of attraction. Finally, we conclude by discussing extensions of these results to spiking models and weight-dependent plasticity.

---

> > ### Author Response · Authors · 2021-08-06
> > **references for the comment above**
> >
> > It looks like some formatting got lost in the Related Work section; I've re-copied it here with formatting fixed. The references from the above sections are also copied below, with journals and years removed due to the character limitation. I referred to the book "Multi-way Array..." by Cichocki et al; that is a chapter in "Nonnegative matrix and tensor factorizations."
> >
> > **Related work and applications**
> >
> > There is a rich literature on generalized or nonlinear forms of Hebbian learning. We briefly discuss the most closely related results, to our knowledge. Learning rules with suitable postsynaptic nonlinearities can allow a neuron to perform independent components analysis (ICA) [40, 41]. These learning rules typically minimize or maximize the kurtosis of the neural response. In contrast, we will show that a simple nonlinear Hebbian model directly finds tensor eigenvectors of a higher-order correlation of the input. Those higher-order input correlations can determine which independent features are learned by gradient-based ICA algorithms [42].
> >
> > Taylor & Coombes showed that a generalization of the Oja rule to higher-order neurons can also learn higher-order correlations [43], which can allow learning independent components [44]. That construction relies on a model with synaptic weights that specifically couple the postsynaptic neuron to coincident inputs; the synaptic weights are themselves a higher-order tensor. We focused on “first-order" synaptic weights with a subsequent nonlinearity in the plasticity.
> >
> > The family of Bienenstock, Cooper & Munro (BCM) learning rules supplement the classic Hebbian model with a stabilizing sliding threshold for potentiation rather than synaptic scaling [45]. BCM rules balance terms driven by third and fourth-order joint moments of the pre- and postsynaptic activity [46]. A triplet STDP model with rate-dependent depression and uncorrelated Poisson spiking has BCM dynamics [47] and can develop selective (sparse) connectivity in response to rate- or correlation-based input patterns [48]. If the input x is drawn from a mixture model then under a BCM rule, the synaptic weights are guaranteed to converge to the class means of the mixture [49].
> >
> > From a computational perspective, nonlinear Hebbian learning has attractive properties for tensor decompositions compared to tensor power iteration [50], which requires the full tensor μ to operate on, with O(Ka+1) space complexity (see eqs. 1, 3 for variable definitions; K is the dimensionality of the inputs). Streaming computations that operate on one input sample at a time, such as those of eq. 2, have O(K) space complexity. The discrete-time dynamics of eq. 1 correspond to streaming tensor power iteration for computing tensor singular vectors [51–53]). With those parameters, eq. 2 can thus be viewed as the continuous-time limiting dynamics of tensor power iteration. Convergence proofs for the discrete-time dynamics require a learning rate that decreases over time. Biological mechanisms for such a scheme are unclear.
> >
> > **References**
> > [1] Hebb DO. The organization of behavior: a neuropsychological theory.
> > [2] Engel A, Van den Broeck C. Statistical Mechanics of Learning.
> > [3] Caporale N, Dan Y. Spike Timing–Dependent Plasticity: A Hebbian Learning Rule.
> > [4] Turrigiano Gina G . The dialectic of Hebb and homeostasis.
> > [5] Zenke F, Gerstner W.Hebbian plasticity requires compensatory processes on multiple timescales.
> > [6] Van Rossum MC, Bi GQ, Turrigiano GG. Stable Hebbian learning from spike timing-dependent plasticity.
> > [7] Rubin J, Lee D, Sompolinsky H. Equilibrium Properties of Temporally Asymmetric Hebbian Plasticity.
> > [8] Bourne JN, Harris KM. Coordination of size and number of excitatory and inhibitory synapses results in a balanced structural plasticity along mature hippocampal CA1 dendrites during LTP.
> > [9] Turrigiano G. Homeostatic Synaptic Plasticity: Local and Global Mechanisms for Stabilizing Neuronal Function.
> > [10] Oja E. Simplified neuron model as a principal component analyzer.
> > [11] Becker S, Plumbley M. Unsupervised neural network learning procedures for feature extraction and classification.
> > [12] Diamantaras KI, Kung SY. Principal component neural networks: theory and applications.
> > [13] Sjöström PJ, Turrigiano GG, Nelson SB. Rate, timing, and cooperativity jointly determine cortical synaptic plasticity.
> > [14] Bi GQ, Wang HX. Temporal asymmetry in spike timing-dependent synaptic plasticity.
> > [15] Froemke RC, Dan Y. Spike-timing-dependent synaptic modification induced by natural spike trains.
> > [16] Wang HX, Gerkin RC, Nauen DW, Bi GQ. Coactivation and timing-dependent integration of synaptic potentiation and depression.
> > [17] Froemke RC, Poo Mm, Dan Y. Spike-timing-dependent synaptic plasticity depends on dendritic location.
> > [18] Froemke RC, Tsay IA, Raad M, Long JD, Dan Y. Contribution of individual spikes in burst- induced long-term synaptic modification.
> > [19] Harvey CD, Svoboda K. Locally dynamic synaptic learning rules in pyramidal neuron dendrites.
> > [20] Harvey CD, Yasuda R, Zhong H, Svoboda K. The spread of Ras activity triggered by Activation of a Single Dendritic Spine.
> > [21] Govindarajan A, Israely I, Huang SY, Tonegawa S. The dendritic branch is the preferred integrative unit for protein synthesis-dependent LTP.
> > [22] Makino H, Malinow R. Compartmentalized versus Global Synaptic Plasticity on Dendrites Controlled by Experience.
> > [23] Kleindienst T, Winnubst J, Roth-Alpermann C, Bonhoeffer T, Lohmann C. Activity-Dependent Clustering of Functional Synaptic Inputs on Developing Hippocampal Dendrites.
> > [24] Chen J, Villa K, Cha J, So PC, Kubota Y, Nedivi E. Clustered Dynamics of Inhibitory Synapses and Dendritic Spines in the Adult Neocortex.
> > [25] Takahashi N, Kitamura K, Matsuo N, Mayford M, Kano M, Matsuki N, et al. Locally Synchronized Synaptic Inputs.
> > [26] Lee KFH, Soares C, Thivierge JP, Béïque JC. Correlated Synaptic Inputs Drive Dendritic Calcium Amplification and Cooperative Plasticity during Clustered Synapse Development.
> > [27] Montani F, Ince RAA, Senatore R, Arabzadeh E, Diamond ME, Panzeri S. The impact of high-order interactions on the rate of synchronous discharge and information trans- mission in somatosensory cortex.
> > [28] Montangie L, Montani F. Effect of interacting second- and third-order stimulus-dependent correlations on population-coding asymmetries.
> > [29] Attneave F. Some informational aspects of visual perception.
> > [30] Barlow HB. Possible principles underlying the transformations of sensory messages.
> > [31] Atick JJ, Redlich AN. Towards a Theory of Early Visual Processing.
> > [32] Atick JJ, Redlich AN. What Does the Retina Know about Natural Scenes?
> > [33] Atick JJ, Redlich AN. Convergent Algorithm for Sensory Receptive Field Development.
> > [34] Dong DW, Atick JJ. Temporal decorrelation: a theory of lagged and nonlagged responses in the lateral geniculate nucleus.
> > [35] Dan Y, Atick JJ, Reid RC. Efficient Coding of Natural Scenes in the Lateral Geniculate Nucleus: Experimental Test of a Computational Theory.
> > [36] Kolda TG, Bader BW. Tensor Decompositions and Applications.
> > [37] Cichocki A, Zdunek R, Phan AH, Amari Si. Nonnegative Matrix and Tensor Factorizations.
> > [38] Cichocki A. Tensor Decompositions: A New Concept in Brain Data Analysis?
> > [39] Williams AH, Kim TH, Wang F, Vyas S, Ryu SI, Shenoy KV, et al. Unsupervised Discovery of Demixed, Low-Dimensional Neural Dynamics across Multiple Timescales through Tensor Component Analysis.
> > [40] Hyvärinen A, Oja E. Simple neuron models for independent component analysis.
> > [41] Bell AJ, Sejnowski TJ. An information-maximization approach to blind separation and blind deconvolution.
> > [42] Lee D, Rokni U, Sompolinsky H. Algorithms for Independent Components Analysis and Higher Order Statistics.
> > [43] Taylor JG, Coombes S. Learning higher order correlations.
> > [44] Ziegaus C, Lang EW. A neural implementation of the JADE algorithm (nJADE) using higher- order neurons.
> > [45] Bienenstock EL, Cooper LN, Munro PW. Theory for the development of neuron selectivity: orientation specificity and binocular interaction in visual cortex.
> > [46] Intrator N, Cooper LN. Objective function formulation of the BCM theory of visual cortical plasticity: Statistical connections, stability conditions.
> > [47] Pfister JP, Gerstner W. Triplets of Spikes in a Model of Spike Timing-Dependent Plasticity.
> > [48] Gjorgjieva J, Clopath C, Audet J, Pfister JP. A triplet spike-timing–dependent plasticity model generalizes the Bienenstock–Cooper–Munro rule to higher-order spatiotemporal cor- relations.
> > [49] Lawlor M, Zucker SW. Feedforward Learning of Mixture Models.
> > [50] De Lathauwer L, De Moor B, Vandewalle J. On the Best Rank-1 and Rank-(R1 ,R2 ,. . .,RN) Approximation of Higher-Order Tensors.
> > [51] Wang Y, Anandkumar A. Online and differentially-private tensor decomposition.
> > [52] Wang PA, Lu CJ. Tensor Decomposition via Simultaneous Power Iteration.
> > [53] Ge R, Huang F, Jin C, Yuan Y. Escaping From Saddle Points — Online Stochastic Gradient for Tensor Decomposition.
> > [54] Qi L. Eigenvalues of a real supersymmetric tensor. Journal of Symbolic Computation.
> > [55] Lim LH. Singular values and eigenvalues of tensors: a variational approach.

---

> > > ### Comment · Reviewer_gbB7 · 2021-08-17
> > > **Response**
> > >
> > > Thank you for the thorough response. The authors have agreed to make changes to the paper, which will help better contextualize.
> > > Although I am not convinced that NeurIPS is the best venue for this paper, I am updating my score to 6.

---

### Official Review · Reviewer_SKSQ · 2021-07-16

**Rating:** 7
**Confidence:** 3

**Summary:**

This paper shows that nonlinear Hebbian rules allow a neuron to learn tensor decompositions of the higher-order input correlation. There have been theoretical works on nonlinear Hebbian learning rules as well as triplet STDP BCM learning rules, but I believe this might be the first that showed systematically and mathematically it enables the learning of tensor eigenvectors and characterize their basins of attractions.

**Ethical Concerns:**

None.

**Limitations And Societal Impact:**

Broader impact section missing.

**Main Review:**

While I understand that higher-order bit,  I found the paper too dense and hard to follow.  I found the math notations in the main text complex, perhaps unnecessarily complex,  and hard to follow.  I looked into the Appendix for clarification but found Eq. 1 in the main text and Eqn. 2 in the Appendix to be inconsistent -- was one of them wrong?  I also could not derive Eq. 3 from Eq. 2 in the Appendix. My puzzle is that even if expanding the norm of J of dt, the 1st order expansion still depends on the norm of vector J.  In contrast,  Eq. 3 in Appendix doesn't depend on the norm of  vector J, but only depend on each element J_j.  The definition of  \dot{J}_j(t) right after Eq. 4 in Appendix is also unclear -- what is the difference between J_i and the bold J_i on the right-hand side of the definition?   While they might be minor issues,  these issues in the first few steps made it sufficiently frustrating for me to press on to evaluate the details. In any case, it is unreasonable for the reviewers to plow through the 20 pages of math in the Appendix.

While I believe the work is probably solid and the contribution fundamental, I wonder if NeurIPS is the appropriate forum. Maybe Neural Computation or Plos Computational Neuroscience would be a better forum. It would be helpful to provide more explanations (e.g. explaining  \mu in Eq. 3 is the input correlation tensor ) and perhaps focus on one or two simple concrete examples to illustrate the results and show what higher-order correlations the neurons actually learn in those examples.  Perhaps something like the long contours in Lawlor and Zucker's mixture model, or perhaps corners or curvatures as in Olshausen's overcomplete sparse coding?    Perhaps they are trying to do this with Figure 1c? But more explanations are necessary.  I don't enjoy the presentation of the paper,  but I am willing to give it a high score because I do think the contribution is potentially important and am willing to give it the benefit of the doubt for now.

**Time Spent Reviewing:**

3 hours.

---

> ### Author Response · Authors · 2021-08-06
> **review response**
>
> Thank you for the review. Our focus in this paper was to study, mostly analytically, the dynamics and convergence of a family of nonlinear Hebbian rules that learn higher-order input features, rather than finding new computational uses for or descriptions of higher-order features. As you point out, a number of studies have examined the use of higher-order features, for example in independent components analysis. Figure 1c shows an example higher-order correlation of the 35x35 pixel natural image patches we used as inputs. Figure 1f does show an example of final synaptic weights.
>
> We have prepared a revised introduction (copied at the end of this response); we hope that this helps motivate the paper. Please let us know if you have any other suggestions for improving its presentation.
>
> Thank you also for pointing out some pain points with the presentation. In the main text, there was indeed a typo in Eq. 1: we left out the factor of $dt / \tau$ that appeared in the supplement. We’ve fixed this.
>
> Throughout the manuscript, we used boldface to denote a vector, matrix or tensor (depending on the variable) and regular type with lower indices to denote elements thereof. We have added a couple sentences before equation 2 describing this. (There were also some missed/switched typefaces in equations, which we’ve fixed.)
>
> The mapping from the discrete weight updates to the limiting differential equation (eq. 1 to eq. 2 in the main text) follows Oja’s original paper (Oja 1982, section 4, which contains a fairly detailed discussion), which we now reference in the supplemental. The basic idea is that for small $dt$,
> $$\vert \vert  \bf{J}(t) + (dt/\tau) \bf{f} \vert \vert_p = 1 + \frac{dt}{\tau} \frac{\partial \vert \vert \bf{J}(t)+(dt / \tau) \bf{f} \vert \vert_p}{\partial dt} \vert_{dt=0} + \mathcal{O} ( dt^2 ).$$
> (Markdown seems to be making everything bold inside the vertical brackets denoting the norm, we apologize that the typeface does not match the description above.)
> In that second term,
> $$\frac{\partial \vert \vert {\bf J}(t)+(dt / \tau) \bf{f} \vert \vert_p}{\partial dt} \vert_{dt=0} = \sum_{j=1}^N J_j \vert J_j \vert^{p-2} f\left(n, \mathrm{x}_j, J_j \right).$$
> This gives rise to the second term in the differential equation for the weights.
>
> In our original submission, the notation $1/ \vert \vert  \bf{J}(t + dt) \vert \vert $ on the right-hand side of Eq. 1 is a slight abuse of notation, since we use $\bf{J}(t+dt)$ on the right-hand side to refer to the pre-normalized values and $J_i(t+dt)$ on the left-hand side to refer to the normalized values. We have switched it so that equation 1 reads $$J_i(t+dt) = \frac{J_i(t) + (dt/\tau) f(n(t), \mathrm{x}_i(t), J_i(t))}{\vert \vert \bf{J}(t) + (dt/\tau) \bf{f}(n(t), \mathrm{\bf{x}} (t), \bf{J}(t)) \vert \vert }$$ .
>
> We have expanded the description of the tensor $\bf{\mu}$, below Eq. 3, to describe it as a higher-order correlation tensor of the inputs.
>
> The revised introduction and related work section follow. The references can be found in the comments under the final review.
>
> **Introduction**
>
> The strength of synaptic connectivity depends on pre- and postsynaptic activity. In Hebbian learning rules, potentiation of the net synaptic weight between two neurons is driven by the correlation between pre- and postsynaptic activity [1]. That postulate is a cornerstone of the theory of synaptic plasticity and learning [2, 3]. In its basic form, the Hebbian model leads to runaway potentiation or depression of synapses, since the pre-post correlation increases with increasing synaptic weight [4]. That runaway potentiation can be stabilized by supplemental homeostatic plasticity dynamics [5], by weight dependence in the learning rule [6, 7], or by synaptic scaling regulating a neuron’s total synaptic weight [8, 9]. In 1982, Erkki Oja observed that a linear neuron with Hebbian plasticity and synaptic scaling learns the first principal component of its inputs [10]. This led to a fountain of research on unsupervised learning in neural networks [11, 12]. The basic Hebbian postulate does not take into account fundamental nonlinear aspects of biological synaptic plasticity in cortical pyramidal neurons. First, synaptic plasticity depends on beyond- pairwise activity correlations [13–18]. Second, spatially clustered and temporally coactive synapses exhibit correlated and cooperative plasticity [19–26]. There is a rich literature on computationally motivated forms of nonlinear Hebbian learning (see 1.1). Here, we will prove that these biologically motivated nonlinearities allow a neuron to learn higher-order features of its inputs.
>
> Principal component analysis (PCA) describes second-order features of a random variable. Both naturalistic stimuli and neural activity can, however, exhibit higher-order correlations—features that are not described by PCA [27, 28]. Canonical models of retinal and thalamic processing whiten inputs, removing pairwise features [29–35]. Beyond-pairwise features, encoded in tensors, can provide a powerful substrate for learning from data [36–39].
>
> **Related work and applications**
>
> There is a rich literature on generalized or nonlinear forms of Hebbian learning. We briefly discuss the most closely related results, to our knowledge. Learning rules with suitable postsynaptic nonlinearities can allow a neuron to perform independent components analysis (ICA) [40, 41]. These learning rules typically minimize or maximize the kurtosis of the neural response. In contrast, we will show that a simple nonlinear Hebbian model directly finds tensor eigenvectors of a higher-order correlation of the input. Those higher-order input correlations can determine which independent features are learned by gradient-based ICA algorithms [42].
>
> Taylor & Coombes showed that a generalization of the Oja rule to higher-order neurons can also learn higher-order correlations [43], which can allow learning independent components [44]. That construction relies on a model with synaptic weights that specifically couple the postsynaptic neuron to coincident inputs; the synaptic weights are themselves a higher-order tensor. We focused on “first-order" synaptic weights with a subsequent nonlinearity in the plasticity.
>
> The family of Bienenstock, Cooper & Munro (BCM) learning rules supplement the classic Hebbian model with a stabilizing sliding threshold for potentiation rather than synaptic scaling [45]. BCM rules balance terms driven by third and fourth-order joint moments of the pre- and postsynaptic activity [46]. A triplet STDP model with rate-dependent depression and uncorrelated Poisson spiking has BCM dynamics [47] and can develop selective (sparse) connectivity in response to rate- or correlation-based input patterns [48]. If the input x is drawn from a mixture model then under a BCM rule, the synaptic weights are guaranteed to converge to the class means of the mixture [49].
>
> From a computational perspective, nonlinear Hebbian learning has attractive properties for tensor decompositions compared to tensor power iteration [50], which requires the full tensor μ to operate on, with O(Ka+1) space complexity (see eqs. 1, 3 for variable definitions; K is the dimensionality of the inputs). Streaming computations that operate on one input sample at a time, such as those of eq. 2, have O(K) space complexity. The discrete-time dynamics of eq. 1 correspond to streaming tensor power iteration for computing tensor singular vectors [51–53]). With those parameters, eq. 2 can thus be viewed as the continuous-time limiting dynamics of tensor power iteration. Convergence proofs for the discrete-time dynamics require a learning rate that decreases over time. Biological mechanisms for such a scheme are unclear.
>
> **Our contribution**
>
> We study the dynamics of a simple family of generalized Hebbian learning rules motivated by these observations discussed above, combined with synaptic scaling (eq. 1). We show that its equilibrium synaptic weights correspond to low-rank tensor decompositions for higher-order input correlations. The order of input correlation (two-point, triplet, etc.) depends on the pre- and postsynaptic nonlinearities of the learning rule. When the only nonlinearity in the plasticity rule is postsynaptic, the steady states are tensor eigenvectors [50, 54, 55] of higher-order input correlations. We prove that the higher-order input correlation tensor’s eigenvectors are attractors of the generalized Hebbian plasticity dynamics and characterize their basins of attraction.
>
> Then, we study further generalizations of these learning rules. We show that any plasticity model (with a finite Taylor expansion in the synaptic input, neural output, and synaptic weight) has steady states that generalize those tensor decompositions to multiple input correlations, including generalized tensor eigenvectors. We show that these generalized tensor eigenvector are stable equilibria of the learning dynamics. Due to the complexity of the arbitrary learning rules, we are unable to fully determine their basins of attraction. We do find that they are contained in an attracting set for the dynamics, and characterize its basin of attraction. Finally, we conclude by discussing extensions of these results to spiking models and weight-dependent plasticity.

---

> > ### Comment · Reviewer_SKSQ · 2021-08-27
> > **Post-rebuttal comment**
> >
> > Thank you for taking the time to address my questions in such details.  I will maintain my "good paper, accept" score.

---

### Official Review · Reviewer_2bLY · 2021-07-16

**Rating:** 7
**Confidence:** 3

**Summary:**

The authors analyze a broad class of nonlinear Hebbian learning rules. These rules generalize the well-studied Oja's learning rule to higher-order input correlations, which can be represented in symmetric higher-order tensors of synaptic input statistics. There is a broad modeling literature on synaptic plasticity, and it is not my primary area of research. Nonetheless, this generalization of Oja's rule appears novel to me, and the extensions outlined in this manuscript feel very natural and interesting.

**Limitations And Societal Impact:**

Limitations are adequately discussed and I foresee no negative societal impacts.

**Main Review:**

The authors analyze a broad class of nonlinear Hebbian learning rules. These rules generalize the well-studied Oja's learning rule to higher-order input correlations, which can be represented in symmetric higher-order tensors of synaptic input statistics. There is a broad modeling literature on synaptic plasticity, and it is not my primary area of research. Nonetheless, this generalization of Oja's rule appears novel to me, and the extensions outlined in this manuscript feel very natural and interesting.

The authors point out that "any plasticity model with a finite Taylor polynomial" can be seen as a special case of their framework. However, there are few insights into this general case, and most of their analysis is confined to simpler cases that are analytically tractable. These special cases are still interesting.

In the classic case of Oja's rule, the largest eigenvector is the unique/global attractor of the system. However, this is not the case for the generalized model considered here. Thus, the authors' results are somewhat pessimistic because the learning dynamics will reach different steady-states depending on the initialization. This makes it more difficult to interpret the functional purpose or outcome of the learning rule, unless one is able to make additional assumptions about the initial state of the system.

On the other hand, it appears that the dynamics do converge, in practice, to the top few eigenvectors when initialized randomly (see fig. 1h). This at least holds in the special cases the authors examined empirically. One of the more useful results in the paper is a proof on the relative sizes of the basins of attraction, which provides good intuition.

One unanswered question I have is whether there are reasons to think the top eigenvectors of higher-order input statistics would be useful features to learn? In Oja's rule, the synaptic weights converge to the top principal component, and thus the neuron becomes tuned to detect salient (i.e. high-variance) features. The interpretation of feature-detectors for higher-order statistics is less intuitive to me, though I know there has been some work done on this -- e.g., https://doi.org/10.1167/13.9.974

Overall, I think these results are very thorough and interesting, if still a bit abstract and general. If the authors could provide readers with more intuition about how higher-order features may be useful features for downstream neural computation, I think it would broaden the appeal and impact of the paper. Either way, I hope to see this paper accepted.

Other comments:

- I attempted to verify all derivations and statements in the supplemental material and I could not find any errors.

- Do the dynamics in fig. 1e actually converge? The trajectories look very volatile.

- The fact that the learning rule does not always converge to the largest eigenvector is not suprising, since recovering the top eigenvector is NP-hard even for symmetric tensors (Hillar & Lim, 2013).

- Notation is somewhat confusing and inconsistent in places. There are three different typefaces for the variable "x" in equations 1 and 2. The variable \mu_{i, \alpha} is boldfaced in equation 3, but is not boldfaced in equation 2. It would be helpful if the explanation that \alpha is a multi-index could be moved a few sentences earlier, right after equation 2.

- On lines 90-91, it is claimed that the first k E-eigenvectors define the best rank-k approximation approximation of \mu in terms of a least-squares criterion. Typically a rank-k approximation is understood to mean a CP-decomposition of the tensor (see Kolda & Bader, 2009), but it seems like here it is meant to be the best rank-k Tucker decomposition to approximate the tensor? This is the sense I got from reading the citation to De Lathauwer (2000), which distriguishes between a rank-R and a rank-(R_1, R_2, ..., R_n) approximation to a tensor. Can the authors clarify these points more explicitly?

- In the caption of Figure 2, a small typo appears: a "v2" should be "v_2"

- On line 553 of the supplement, a reference / citation is broken.

**Time Spent Reviewing:**

5

---

> ### Author Response · Authors · 2021-08-06
> **review response**
>
> Thank you for your thorough review and your support. We especially appreciate the time you spent to go through the supplemental material.
>
> We have made several revisions to our manuscript in response to your and the other reviewers’ comments. In particular, we have prepared a revised introduction and “related work” section, which we copy below. The introduction now includes a paragraph discussing applications of tensor decompositions, which we hope helps motivate the work. We are open to moving the related work section into the introduction as below (or otherwise revising it) if the reviewers think that would improve the manuscript.
>
> In fig. 1e, the dynamics are stochastic due to the finite timescale ($dt / \tau$), which generates fluctuations around the attractors we described. The magnitude of those fluctuations scales as  (dt / \tau)^2.
>
> We now include the Hillar & Lim reference to help motivate our analytical investigation of the dynamics (before theorem 1).
>
> Regarding notation: thank you for pointing this out. We have fixed those mismatched typefaces for the inputs $\bf{\mathrm{x}}$, and gone through and checked it in the rest of the paper. We have also added a couple sentences before equation 2 regarding the use of boldface notation for vectors/matrices/tensors, and lower indices with regular type for elements thereof.
>
> Lines 90-91: Thank you for catching this! As you say, the top $n$ E-eigenvectors are the components of the best rank-$n$ Tucker decomposition of a symmetric tensor (De Lathauwer et al. 2000, A multi-linear...), but the components of the Tucker decomposition are indeed not the best low-rank approximation of a tensor. We have clarified that paragraph:
>
> "
> The E-eigenvectors correspond to the components of the Tucker decomposition of $\bf{\mu}$, providing a low-rank approximation of $\bf{\mu}$ (Tucker 1966, De Lathauwer et al. 2000 (On the best..)). (The E-eigenvalues $\lambda$ are the norm of sub-tensors of the core tensor in the Tucker decomposition.)
>  With no weight-dependence ($c=0$), steady states of the nonlinear Hebbian dynamics can thus allow a low-rank approximation of a higher-order input correlation. In the remainder of the paper, we will usually focus on parameter sets with $c=0$, and use ``tensor eigenvector" to refer to those of eq. 5.
> "
>
> Caption of Figure 2 typo: thank you for pointing this out, we’ve fixed it.
>
> Broken citation at line 553 of the supplement: thank you for pointing this out, we’ve fixed it.
>
> The revised introduction and related work sections follow. The references can be found under the comments of reviewer 4.
>
> **Introduction**
>
> The strength of synaptic connectivity depends on pre- and postsynaptic activity. In Hebbian learning rules, potentiation of the net synaptic weight between two neurons is driven by the correlation between pre- and postsynaptic activity [1]. That postulate is a cornerstone of the theory of synaptic plasticity and learning [2, 3]. In its basic form, the Hebbian model leads to runaway potentiation or depression of synapses, since the pre-post correlation increases with increasing synaptic weight [4]. That runaway potentiation can be stabilized by supplemental homeostatic plasticity dynamics [5], by weight dependence in the learning rule [6, 7], or by synaptic scaling regulating a neuron’s total synaptic weight [8, 9]. In 1982, Erkki Oja observed that a linear neuron with Hebbian plasticity and synaptic scaling learns the first principal component of its inputs [10]. This led to a fountain of research on unsupervised learning in neural networks [11, 12]. The basic Hebbian postulate does not take into account fundamental nonlinear aspects of biological synaptic plasticity in cortical pyramidal neurons. First, synaptic plasticity depends on beyond- pairwise activity correlations [13–18]. Second, spatially clustered and temporally coactive synapses exhibit correlated and cooperative plasticity [19–26]. There is a rich literature on computationally motivated forms of nonlinear Hebbian learning (see 1.1). Here, we will prove that these biologically motivated nonlinearities allow a neuron to learn higher-order features of its inputs.
>
> Principal component analysis (PCA) describes second-order features of a random variable. Both naturalistic stimuli and neural activity can, however, exhibit higher-order correlations—features that are not described by PCA [27, 28]. Canonical models of retinal and thalamic processing whiten inputs, removing pairwise features [29–35]. Beyond-pairwise features, encoded in tensors, can provide a powerful substrate for learning from data [36–39].
>
> **Related work and applications**
>
> There is a rich literature on generalized or nonlinear forms of Hebbian learning. We briefly discuss the most closely related results, to our knowledge. Learning rules with suitable postsynaptic nonlinearities can allow a neuron to perform independent components analysis (ICA) [40, 41]. These learning rules typically minimize or maximize the kurtosis of the neural response. In contrast, we will show that a simple nonlinear Hebbian model directly finds tensor eigenvectors of a higher-order correlation of the input. Those higher-order input correlations can determine which independent features are learned by gradient-based ICA algorithms [42].
>
> Taylor & Coombes showed that a generalization of the Oja rule to higher-order neurons can also learn higher-order correlations [43], which can allow learning independent components [44]. That construction relies on a model with synaptic weights that specifically couple the postsynaptic neuron to coincident inputs; the synaptic weights are themselves a higher-order tensor. We focused on “first-order" synaptic weights with a subsequent nonlinearity in the plasticity.
>
> The family of Bienenstock, Cooper & Munro (BCM) learning rules supplement the classic Hebbian model with a stabilizing sliding threshold for potentiation rather than synaptic scaling [45]. BCM rules balance terms driven by third and fourth-order joint moments of the pre- and postsynaptic activity [46]. A triplet STDP model with rate-dependent depression and uncorrelated Poisson spiking has BCM dynamics [47] and can develop selective (sparse) connectivity in response to rate- or correlation-based input patterns [48]. If the input x is drawn from a mixture model then under a BCM rule, the synaptic weights are guaranteed to converge to the class means of the mixture [49].
>
> From a computational perspective, nonlinear Hebbian learning has attractive properties for tensor decompositions compared to tensor power iteration [50], which requires the full tensor μ to operate on, with O(Ka+1) space complexity (see eqs. 1, 3 for variable definitions; K is the dimensionality of the inputs). Streaming computations that operate on one input sample at a time, such as those of eq. 2, have O(K) space complexity. The discrete-time dynamics of eq. 1 correspond to streaming tensor power iteration for computing tensor singular vectors [51–53]). With those parameters, eq. 2 can thus be viewed as the continuous-time limiting dynamics of tensor power iteration. Convergence proofs for the discrete-time dynamics require a learning rate that decreases over time. Biological mechanisms for such a scheme are unclear.
>
> **Our contribution**
>
> We study the dynamics of a simple family of generalized Hebbian learning rules motivated by these observations discussed above, combined with synaptic scaling (eq. 1). We show that its equilibrium synaptic weights correspond to low-rank tensor decompositions for higher-order input correlations. The order of input correlation (two-point, triplet, etc.) depends on the pre- and postsynaptic nonlinearities of the learning rule. When the only nonlinearity in the plasticity rule is postsynaptic, the steady states are tensor eigenvectors [50, 54, 55] of higher-order input correlations. We prove that the higher-order input correlation tensor’s eigenvectors are attractors of the generalized Hebbian plasticity dynamics and characterize their basins of attraction.
>
> Then, we study further generalizations of these learning rules. We show that any plasticity model (with a finite Taylor expansion in the synaptic input, neural output, and synaptic weight) has steady states that generalize those tensor decompositions to multiple input correlations, including generalized tensor eigenvectors. We show that these generalized tensor eigenvector are stable equilibria of the learning dynamics. Due to the complexity of the arbitrary learning rules, we are unable to fully determine their basins of attraction. We do find that they are contained in an attracting set for the dynamics, and characterize its basin of attraction. Finally, we conclude by discussing extensions of these results to spiking models and weight-dependent plasticity.

---

### Official Review · Reviewer_aWnj · 2021-07-21

**Rating:** 9
**Confidence:** 3

**Summary:**

The authors study generalized, nonlinear Hebbian learning rules, and find that neurons with such plasticity and feedforward inputs can learn tensor decompositions of higher-order correlations.  They further demonstrate that for simple, biologically plausible scenarios, the tensor eigenvectors are attractors. After determining the basins of attraction, the authors show that the basin of the dominant eigenvector has the largest volume.  Finally, they show that any plasticity rule with a finite Taylor expansion in synaptic input, neural output, and weight variables has equilibrium states  that are similar to those already found and describe higher-order input correlations.

**Limitations And Societal Impact:**

Yes

**Main Review:**

The authors have extended a large body of prior work on local linear learning rules. These Hebbian (and generalized Hebbian) rules have been shown to allow neurons to model input statistics; for example, Oja's rule has been shown to be stable and instantiate an online algorithm for PCA.  The thorough analysis encapsulated in this manuscript represents a welcome advance in this classical unsupervised learning framework.

**Time Spent Reviewing:**

2.5

---

> ### Author Response · Authors · 2021-08-06
> **review response**
>
> Thank you for your support! We’re very glad you enjoyed our paper and appreciated the approach. We're grateful for the time you spent reviewing it.

---

### Decision · Program_Chairs · 2021-09-27

**Decision:**

Accept (Poster)

**Comment:**

This paper shows how nonlinear Hebbian rules learn tensor decompositions of higher-order input correlations. This is an important contribution to the literature on Hebbian synaptic plasticity. Original submission missed important citations and related work, however the revised introduction proposed by the authors better frame the work and its contributions.